

# 3 **Investigating basin-scale water budget dynamics in 18 rivers across**

# 4 **Tibetan Plateau through multiple datasets**

Wenbin Liu[a], Fubao Sun[a,b*], Yanzhong Li[a], Guoqing Zhang[c,d], Yan-Fang Sang[a],
Wee Ho Lim[a,e], Jiahong Liu[f], Hong Wang[a] ,Peng Bai[a]
[a] Key Laboratory of Water Cycle and Related Land Surface Processes, Institute of Geographic
Sciences and Natural Resources Research, Chinese Academy of Sciences, Beijing 100101, China
[b] Hexi University, Zhangye 734000, China [c] Key Laboratory of Tibetan Environmental Changes
and Land Surface Processes, Institute of Tibetan Plateau Research, Chinese Academy of Sciences,
Beijing 100101, China [d] CAS Center for Excellent in Tibetan Plateau Earth Sciences, Beijing
100101, China [e] Environmental Change Institute, Oxford University Centre for the Environment,
School of Geography and the Environment, University of Oxford , Oxford OX1 3QY, UK [f] Key
Laboratory of Simulation and Regulation of Water Cycle in River Basin, China Institute of Water
Resources and Hydropower Research, Beijing 100038, China
**Corresponding Author**: Dr. Fubao Sun (Sunfb@igsnrr.ac.cn), Key Laboratory of Water Cycle
and Related Land Surface Processes, Institute of Geographic Sciences and Natural Resources
Research, Chinese Academy of Sciences



**Abstract** The dynamics of basin-scale water budgets are not well understood
nowadays over the Tibetan Plateau (TP) due to the lack of hydro-climatic
observations. In this study, we investigate seasonal cycles and trends of water budget
components (e.g., precipitation-P, evapotranspiration-ET and runoff-Q) in eighteen TP
river basins during the period 1982-2011 through the use of multi-source datasets (e.g.,
in situ observations, satellite retrievals, reanalysis outputs and land surface model
simulations). A water balance-based two-step procedure, which considers the changes
in basin-scale water storage at the annual scale, is also adopted to calculate actual ET.
The results indicated that precipitation (mainly snowfall from mid-autumn to next
spring), which mainly concentrated during June-October (varied among different
monsoons-impacted basins), was the major contributor to the runoff in TP basins.
Increased P, ET and Q were found in most TP basins during the past 30 years except
for the upper Yellow River basin and some sub-basins of Yalong River, which were
mainly affected by the weakening East Asian Monsoon. Moreover, the aridity index
(PET/P) and runoff coefficient (Q/P) decreased in most basins, which were in
agreement with the warming and moistening climate in the Tibetan Plateau. The
results obtained demonstrated the usefulness of integrating multi-source datasets to
hydrological applications in the data-sparse regions. More generally, such approach
might offer helpful insights towards understanding the water and energy budgets and
sustainability of water resource management practices of data-sparse regions in a
changing environment.



## 1 Introduction

As the highest plateau in the globe (the average elevation is higher than 4000 meters

above the sea level), the Tibetan Plateau (TP, also called "the roof of the world" or

"the third Pole") is regarded as one of the most vulnerable regions under a warming

climate and is exposed to strong interactions among atmosphere, hydrosphere,

biosphere and cryosphere in the earth system (Duan and Wu, 2006; Yao et al., 2012;

Liu et al., 2016b). It also serves as the "Asian water tower" from which some major

Asian rivers such as Yellow River, Yangtze River, Brahmaputra River, Mekong River,

Indus River, etc., originate, which is a vital water resource to support the livelihood of

hundreds of millions of people in China and the neighboring Asian countries

(Immerzeel et al., 2010; Zhang et al., 2013). Hence sound knowledge of water budget

and hydrological regimes in TP river basins and its response to the changing

environment would have practical relevance for achieving sustainable water resource

management and environmental protection in this part of the world (Yang et al., 2014;

Chen et al., 2015).

Despite the importance of TP in this geographic region, advance in hydrological and

land surfaces studies in this region has been limited by data scarcity (Zhang et al.,

2007; Li F. et al., 2013; Liu X. et al., 2016). For instance, less than 80 observation

stations (~10% of a total of ~750 observation station across China) have been

established in TP by the Chinese Meteorological Administration (CMA) since the

mid-20[th] century (Wang and Zeng, 2012). These stations are generally sparse and

unevenly distributed at relatively low elevation regions, focus only on the

meteorological variables and lack of other land surface observations such as

evapotranspiration, snow water equivalent and latent heat fluxes. In addition,





long-term observations of river discharge, snow depth, lake depth and glacier melts in
the TP are also absent (Akhta et al., 2009; Ma et al., 2016). Therefore, the water
budget and hydrological regimes for each river basin of TP and their relation with
atmospheric circulations are poorly understood (Cuo et al., 2014; Xu et al., 2016).
Whilst this shortcoming could be resolved through installation of in-situ monitoring
systems (Yang et al., 2013; Zhou et al., 2013; Ma et al., 2015), the overall cost of
running the operational sites would be substantial. Another workaround would be
through modeling approach, i.e., feeding remote sensing information and
meteorological forcing data into physically-based land surface model (LSM) to
simulate the basin-wide water budget (Bookhagen and Burbank, 2010; Xue et al.,
2013; Zhang et al., 2013; Cuo et al., 2015; Zhou et al., 2015; Wang et al., 2016).
However, such approach is not immune from the issue of data scarcity at multiple
river basins (with varied sizes and/or terrain complexities) for supporting model
calibration and validation purposes (Li F. et al., 2014).

Most recently, several global (or regional) datasets relevant to the calculation of water
budget have been released. They include remote sensing-based retrievals (Tapley et al.,
2004; Zhang et al., 2010; Long et al., 2014; Zhang Y. et al., 2016), land surface model
(LSM) simulations (Rui, 2011), reanalysis outputs (Berrisford et al., 2011; Kobayashi
et al., 2015) and gridded forcing data interpolated from the in situ observations
(Harris et al., 2014). For example, there are many products related to terrestrial
evapotranspiration (ET) such as GLEAM_E (Global Land surface Evaporation: the
Amsterdam Methodology, Miralles et al., 2011a), MTE_E (a product integrated the
point-wise ET observation at FLUXNET sites with geospatial information extracted
from surface meteorological observations and remote sensing in a machine-leaning



algorithm, Jung et al., 2010 ), LSM-simulated ETs from Global Land Data
Assimilation System version 2 (GLDAS-2) with different land surface schemes
(Rodell et al., 2004), ETs from Japanese 55-year reanalysis (JRA55_E), the
ERA-Interim global atmospheric reanalysis dataset (ERAI_E) and the National
Aeronautic and Space Administration (NASA) Modern Era Retrosphective-analysis
for Research and Application (MERRA) reanalysis data (Lucchesi, 2012). Moreover,
there are also several global or regional LSM-based runoff simulations from GLDAS
and the Variable Infiltration Capacity (VIC) model (Zhang et al., 2014). A few
attempts have been made to validate multiple datasets for certain water budget
components and to explore their possible hydrological implications. For example, Li
X. et al. (2014) and Liu et al. (2016a) evaluated multiple ET estimates against the
water balance method at annual and monthly time scales. Bai et al. (2016) assessed
streamflow simulations of GLDAS LSMs in five major rivers over the TP based on
the discharge observations. Although uncertainties might exist among different
datasets with various spatial and temporal resolutions and calculated using different
algorithms (Xia et al., 2012), they offer an opportunity to examine the general
basin-wide water budgets and their uncertainties in gauge-sparse regions such as the
TP considered in this study.

From the multiple datasets perspective, this study aims to investigate the water budget
in 18 TP river basins distributed across the Tibetan Plateau; and evaluate seasonal
cycles and annual trends of these water budget components. This paper is organized
as follows: the datasets and methods applied in this study are described in Sect.2. The
results of season cycles and annual trends of water budget components for the river
basins are presented and discussed in Sect.3. The uncertainties arise from employing





multiple datasets are also discussed in the same section. In Sect.4, we generalize our
findings which would be helpful for understanding the water balances of the river
basins under constant influence of interplay between westerlies and monsoons (e.g.,
Indian monsoon, East Asian monsoon) in the Tibetan Plateau.

**2   Data and methods**
**2.1 Multiple datasets used**
**2.1.1 Runoff, precipitation and terrestrial storage change**
We obtained the observed daily runoff (Q) during the period 1982-2011 from the
National Hydrology Almanac of China (Table 1). There are < 30% missing data in
some gauging stations such as Yajiang, Tongren, Gandatan and Zelingou. Therefore,
the VIC Retrospective Land Surface Dataset over China (1952~2012, VIC_IGSNRR
simulated) with a spatial resolution of 0.25 degree and a daily temporal resolution
from the Geographic Sciences and Natural Resources Research (IGSNRR), Chinese
Academy of Sciences, is also used. This dataset is derived from the VIC model forced
by the gridded daily observed meteorological forcing (IGSNRR_forcing) (Zhang et al.,
2014). A degree-day scheme was used in the model to account for the influences of
snow and glacier on hydrological processes.

In terms of precipitation (P), we used the gridded monthly precipitation dataset
available at CMA (spatial resolution of 0.5 degree; 1961-2011; interpolated from
observations of 2372 national meteorological stations using the Thin Plate Spline
method) (Table 1). Since the reliability of this dataset might be restricted by the
relatively sparse stations and complex terrain conditions of TP, we make an attempt to
incorporate two other precipitation datasets ((IGSNRR_forcing and Tropical Rainfall



Measuring Mission TRMM 3B43 V7). The precipitation from IGSNRR forcing
datasets (0.25 degree) was derived by interpolating gauged daily precipitation from
756 CMA stations based on the synergraphic mapping system algorithm (Shepard,
1984; Zhang et al., 2014) and was further bias-corrected using the CMA gridded
precipitation.

To get the change in terrestrial storage (ΔS), we used three latest global terrestrial
water storage anomaly and water storage change datasets (available on the GRACE
Tellus website: http://grace.jpl.nasa.gov/) that were retrieved from the Gravity
Recovery and Climate Experiment (GRACE, Tapley et al., 2004; Landerer and
Swenson, 2012; Long et al., 2014). Briefly, they were processed separately at the Jet
Propulsion Laboratory (JPL), the GeoForschungsZentrum (GFZ) and the Center for
Space Research at the University of Texas (CSR). To minimize the errors and
uncertainty of extracted ΔS, we averaged these GRACE retrievals (2002-2013) from
different processing centers in this study.

**2.1.2 Temperature, potential evaporation and ET**
We obtained the monthly gridded temperature dataset (0.5 degree) from CMA; and
potential evaporation (PET) dataset (0.5 degree, Harris et al., 2013) from Climatic
Research Unit (CRU), University of East Anglia. Moreover, we used six global
/regional ET products (four diagnostic products and two LSMs simulations, Table 2),
namely (1) GLEAM_E (Miralles et al., 2010, 2011), which consists of three sources
of ET (transpiration, soil evaporation and interception) for bare soil, short vegetation
and vegetation with a tall canopy calculated using a set of algorithm (www.gleam.eu),



(2) GNoah_E simulated using GLDAS-2 with the Catchment Noah scheme
(http://disc.sci.gsfc.nasa.gove/hydrology/data-holdings) (Rodell et al., 2004), (3)
Zhang_E (Zhang et al., 2010), which is estimated using the modified
Penman-Monteith equation forced with MODIS data, satellite-based vegetation
parameters and meteorological observations (http://www.ntsg.umt.edu/project/et), (4)
MET_E (Jung et al., 2010) (https://www.bgc-jena.mpg.de/geodb/projects/Home.phs),
(5) VIC_E (Zhang et al., 2014) from VIC_IGSNRR simulations
(http://hydro.igsnrr.ac.cn/public/vic_outputs.html) and (6) PML_E (Zhang Y. et al.,
2016) computed from global observation-driven Penman-Monteith-Leuning (PML)
model (https://data.csiro.au/dap/landingpage?pid=csiro:17375&v=2&d=true).

**2.1.3 Vegetation and snow/glacier parameters**
To quantify the dynamics of vegetation of each river basin, we applied the
Normalized Difference Vegetation Index (NDVI) and the Leaf Area Index (LAI)
(Table 1). Briefly, the NDVI data was obtained from the Global Inventory Modeling
and Mapping Studies (GIMMS) (Turker et al., 2005)
(https://nex.nasa.gov/nex/projects/1349/wiki/general_data_description_and_access/)
while the LAI data was collected from the Global Land Surface Satellite (GLASS)
products (http://www.glcf.umd.edu/data/lai/) (Liang and Xiao, 2012). Whist the
change in seasonal snow cover and glacier has significant impact on the water and
energy budgets in TP river basins; it remains a technical challenge to get reliable
observations due to harsh environment (especially at the basin scale). However,
recently available satellite-based/LSM-simulated products might provide adequate
characterization of the variation of snow cover and glacier. To quantify the change in



snow cover at each basin, we applied the daily cloud free snow composite product
from MODIS Terra-Aqua and the Interactive Multisensor Snow and Ice Mapping
System for the Tibetan Plateau (Zhang et al., 2012; Yu et al., 2015), in conjunction
with the snow water equivalent (SWE) retrieved from Global Snow Monitoring for
Climate Research product (GlobSnow-2, http://www.globsnow.info/) and the
VIC_IGSNRR simulations (Takala et al., 2011; Zhang et al., 2014). We extracted
general distribution of glacier of TP from the Second Glacier Inventory Dataset of
China (Guo et al., 2014). All gridded datasets used were first uniformly interpolated to
a spatial resolution of 0.5 degree based on the bilinear interpolation to make their
inter-comparison possible. The datasets were then extracted for each of TP basins.

**2.1.4 Monsoon indices**
In general, the TP climate is under the influences of the westerlies, Indian summer
monsoon and East Asian summer monsoon (Yao et al., 2012). To investigate the
changes of monsoon systems and their potential impacts on water budgets in the TP
basins, we used three monsoon indices, namely Asian Zonal Circulation Index (AZCI),
Indian Ocean Dipole Mode Index (IODMI) and East Asian Summer Monsoon Index
(EASMI). Briefly, the IODMI (reflects the dynamics of Indian Summer Monsoon) is
an indicator of the east-west temperature gradient across the tropical Indian Ocean
(Saji et al., 1999), which can be downloaded from the following website:
http://www.jamstec.go.jp/frcgc/research/d1/iod/HTML/Dipole%20Mode%20Index.ht
ml. The EASMI and AZCI ($60^{o}$-$150^{o}$E) reflect the dynamics of East Asian summer
monsoon (Li and Zeng, 2002) and the westerlies (represented by Asian Zonal
Circulation index), which can be obtained from Beijing Normal University
(http://ljp.gcess.cn/dct/page/65577) and the National Climate Center of China



(http://ncc.cma.gov.cn/Website/index.php?ChannelID=43WCHID=5), respectively.

**2.1.5 Study basins**
In this study, we selected 18 river basins of varied sizes (range: 2832-191235 km$^2$;
see Table 1 for details) with adequate runoff data over a 30-year period (1982-2011).
They are distributed in the northwestern, southeastern and eastern parts of the plateau
with multiyear-mean and basin-averaged temperature and precipitation ranging from
-5.68 to 0.97 $^{\circ}$C and 128 to 717 mm, which are solely dominated or under the
combined influences of the westerlies, the Indian Summer monsoon and the East
Asian monsoon (Yao et al., 2012). There are more glacier and snow covers in the
westerlies-dominant basins such as Yerqiang, Yulongkashi and Keliya (10.86~23.27%
and 29.16~35.95%, respectively); less for the East Asian monsoon-dominated basins
such as Yellow, Yangtze and Bayin (0~0.96% and 9.42~20.05%, respectively) (Table

232    2).




**2.2 Methods**
**2.2.1 Water balance-based ET estimation**
The basin-wide water balance at the monthly and annual timescales could be written
as the principle of mass conservation (also known as the continuity equation, Oliveira
et al., 2014) of basin-wide precipitation (P, mm), evapotranspiration (ET$_{wb}$, mm),
runoff (Q, mm) as well as terrestrial water storage change (ΔS, mm),
$$ET_{wb} = P - Q - \Delta S \qquad (1)$$
The terrestrial water storage (ΔS) in Eq. (1) includes the surface, subsurface and



ground water changes. It has been demonstrated cannot be neglected in water balance
calculation over monthly and annual timescales due to snow cover change and
anthropogenic interferences (e.g., reservoir operation, agricultural water withdrawal)
(Liu et al., 2016a). For the period 2002-2011, we calculated basin-wide ET ($ET_{wb}$)
directly using the GRACE-derived $\Delta S$ in Eq. (1). Since GRACE data is absent
before 2002, we calculated the monthly $ET_{wb}$ using the following two-step
bias-correction procedure (Li X. et al., 2014). We defined $P - Q$ in Eq. (1) as biased
ET ($ET_{biased}$, available from 1982 to 2011) relative to the "true" ET ($ET_{wb} = P - Q -$
$\Delta S$, available during the period 2002-2011 when the GRACE data is available). Over
the period 2002-2011, we first fitted $ET_{biased}$ and $ET_{wb}$ series separately using
different gamma distributions, which has been evidenced as an proper method for
modeling the probability distribution of ET (Bouraoui et al., 1999). The monthly
$ET_{biased}$ series (2002-2011) can then be bias-corrected through the inverse function
($F^{-1}$) of the gamma cumulative distribution function (CDF, F) of $ET_{wb}$ by matching
the cumulative probabilities between two CDFs as follow (Liu et al., 2016a),
$$ET_{corrrected}(m) = F^{-1}(F(ET_{biased}(m)|\alpha_{biased}, \beta_{biased})|\alpha_{wb}, \beta_{wb}) \qquad (2)$$
Here $\alpha_{biased}, \beta_{biased}$ and $\alpha_{wb}, \beta_{wb}$ are shape and scale parameters of
gamma distributions for $ET_{biased}$ and $ET_{wb}$. $ET_{corrected}(m)$ and $ET_{biased}(m)$
represent the monthly corrected and biased ET, respectively. The bias correction
procedure can be flexibly applied to the period 1983-2011 by matching the CDF
of $ET_{biased}$ (1983-2011) to that of $ET_{corrected}$ (2002-2011). The second step of
bias correction is to eliminate the annual bias through the ratio of annual
$ET_{biased}$ to annual $ET_{corrected}$ calculated in the first step using the following
method,





$$ET_{final}(m) = \frac{ET_{biased}(a)}{ET_{corrected}(a)} \times ET_{corrected}(m) \qquad (3)$$

where $ET_{final}(m)$ is the final monthly ET after bias correction. $ET_{biased}(a)$ and
$ET_{corrected}(a)$ represent the annual biased and corrected ET while
$ET_{corrected}(m)$ is the monthly corrected ET obtained from the first step. The
procedure was then applied to correct the monthly $ET_{biased}$ series and
calculated the monthly $ET_{corrected}$ during the period 1982-2001 for all TP
basins. We take these results as sufficient representation of the "true" ET ($ET_{wb}$)
for evaluating multiple ET products and trend analysis. "

**2.2.2 Modified Mann-Kendall test method**
The Mann-Kendall (MK) test is a rank-based nonparametric approach which is less
sensitive to outlier relative to other parametric statistics, but it is sometimes
influenced by the serial correlation of time series. Pre-whitening is often used to
eliminate the influence of lag-1 autocorrelation before the use of MK test. For
example, $X$ $(X_1, X_2, ..., X_n)$ is a time series data, it will be replaced by $(X_2 -$
$cX_1, X_3 - cX_2, ..., X_{n+1} - cX_n)$ in pre-whitening if the lag-1 autocorrelation
coefficient (c) is larger than 0.1 (von Storch, 1995). However, significant lag-i
autocorrelation may still be detected after pre-whitening because only the lag-1
autocorrelation is considered in pre-whitening (Zhang et al., 2013). Moreover, it
sometimes underestimate the trend for a given time series (Yue et al., 2002). Hamed
and Rao (1998) proposed a modified version of MK test (MMK) to consider the lag-i
autocorrelation and related robustness of the autocorrelation through the use of
equivalent sample size, which has been widely used in previous studies during the last
five decades (McVicar et al., 2012; Zhang et al., 2013; Liu and Sun, 2016). In the
MMK approach, if the lag-i autocorrelation coefficients are significantly distinct from





zero, the original variance of MK statistics will be replaced by the modified one. In
this study, we used the MMK approach to quantify the trends of water budget
components in18 TP basins and the significance of trend was tested at the >95%
confidence level.

**2.2.3 Uncertainty analysis**
The uncertainty associated multi-source dataset used (no observation or the
observations are not adequate at the basin scale) for quantifying the dynamics of
certain water budget components (i.e., ET and precipitation) are also analyzed. We
investigate the seasonal cycles and trends of these components by using different
datasets together in the analysis to show the potential uncertainties in this study.

**3    Results and Discussion**
**3.1 ET evaluation and General hydrological characteristics of 18 TP basins**
We first assessed the VIC_IGSNRR simulated runoff against the observations for
each basin (for example, at Tangnaihai and Pangduo stations in Fig.2). If the Nash
Efficiency coefficient (NSE) between the observation and simulation is above 0.65,
the VIC_IGSNRR simulated runoff is acceptable and could be used to replace the
missing runoff values for a given basin. Moreover, the CMA precipitation is
consistent with TRMM (Corr = 0.86, RMSE = 8.34 mm/month) and IGSNRR forcing
(Corr = 0.94, RMSE = 7.15mm/month) precipitation for multiple basins (i.e., for the
smallest basin above Tongren station, Fig.2). Moreover, the magnitudes of
GRACE-derived annual mean water storage change ($\Delta S$) in 18 TP basins are
relatively less than those for other water balance components such as annual P, Q and
ET (Table 3). The uncertainties among GRACE-derived annual mean  $\Delta S$  from



different data processing centers (CSR, GFZ and JPL) are small for 18 basins except
for the basins controlled by Gadatan and Tangnaihai stations.


We then evaluated six ET products in 18 TP basins against our calculated $ET_{wb}$ at a
monthly basis during the period 1983-2006 (Fig. 3). The ranges of monthly averaged
ET among different basins (approximately 4−39 mm month$^{-1}$) are very close for all
products compare to that calculated from the $ET_{wb}$(6−42 mm month$^{-1}$). However,
GLEAM_E (correlation coefficient: Corr = 0.85 and root-mean-square-error: RMSE =
5.69 mm month$^{-1}$) and VIC_E (Corr = 0.82 and RMSE = 6.16 mm month$^{-1}$) perform
relatively better than others. Although Zhang_E and GNoah_E were found closely
correlated to monthly $ET_{wb}$ in the upper Yellow River, the upper Yangtze River,
Qiangtang and Qaidam basins (Li X. et al., 2014), they did not exhibit overall good
performances (Corr = 0.61, RMSE = 7.97 mm month$^{-1}$ for Zhang_E and Corr = 0.42,
RMSE = 10.16 mm month$^{-1}$ for GNoah_E) for 18 TP basin used in this study. We thus
use GLEAM_E and VIC_E together with $ET_{wb}$ to anayize the seasonal cycles and
trends of ET in 18 TP basins in the following sections.

To investigate the general hydroclimatic characteristics of river basins over the TP, we
classify 18 basins into three categories, namely westerlies-dominated basins
(Yerqiang, Yulongkashi and Kelia), Indian monsoon-dominated basins (Brahmaputra
and Salween), and East Asian monsoon-dominated basins (Yellow, Yalong and
Yangtze) referred to Tian et al. (2007), Yao et al. (2012) and Dong et al. (2016).
Interestingly, they are clustered into three groups under Budyko framework (Budyko,
1974; Zhang D. et al., 2016) with relatively lower evaporative index in Indian





monsoon-dominant basins and higher aridity index in westerlies-dominant basins,
which reveal various long-term hydroclimatologic conditions (Fig. 4). Overall, the
annual mean air temperature increases (-5.68 ~0.97 $^{o}$C) while multiyear mean glacier
area (and thus the glacier melt normalized by precipitation) decreases (23.27 ~ 0%)
gradually from the westerlies-dominant, Indian monsoon-dominant to East Asian
monsoon-dominant basins. The vegetation status (NDVI range: 0.05~0.43; LAI range:
0.03~0.83) tends to be better and ET increases (and thus runoff coefficient gradually
decreases) from cold to warm basins (Fig. 4 and Table 1). The $R^2$ between
basin-averaged NDVI and ET is 0.76 which shows a clear vegetation control on ET in
18 TP basins. The results are in line with Shen et al. (2015), which indicated that the
spatial pattern of ET trend was significantly and positively correlated with NDVI
trend over the TP. The dominant climate systems are overall discrepant for the three
TP regions with different water-energy characteristics and sources of water vapor. The
westerlies-controlled basins are relatively colder than the Indian monsoon-dominated
basins, thus they develop more glaciers (and thus have more snow melt contributions
to total river streamflow) and have relatively less vegetation (and thus limit vegetation
transpiration). It is a general picture of hydrological regime in high-altitude and cold
regions (Zhang et al., 2013; Cuo et al., 2014), which could be interpreted from the
perspective of multi-source datasets in the data-sparse TP.

**3.2 Seasonal cycles of basin-wide water budget components for the TP basins**
The multi-year means of water budget components (i.e., P, Q, ET, snow cover and
SWE) and vegetation parameters (i.e., NDVI and LAI) are calculated for each
calendar month and for 18 TP river basins using multi-source datasets available from
1982 to 2011. Overall, the seasonal variations of P, Q, ET, air temperature and





vegetation parameters are similar in all TP basins with peak values occurred in May to
September (Fig.5 and Fig.6). The seasonal cycles of snow cover and SWE are
generally consistent among the basins (the peak values mainly occur from October to
next April, Fig.7). With the ascending air temperature from cold to warm months, the
basin-wide precipitation increases and vegetation cover expands gradually (the
basin-wide ET also increase). Meanwhile, snow cover and glaciers retreat gradually
with the melt water supplying the river discharge together with precipitation. The
inter-basin variations of hydrological regime are to a large extent linked to the climate
systems that prevail over the TP.

Although the temporal patterns of hydrological components are generally analogous,
they vary among the parameters, climate zones and even basins (Zhou et al., 2005).
For example, relative to air temperature, the seasonal pattern of runoff is similar to
precipitation which reveals that runoff is mainly controlled by precipitation in most
TP basins. It is in agreement with that summarized by Cuo et al. (2014). In the
westerlies-dominated basins, the peak values of precipitation and runoff mainly
concentrate in June-August, which contribute approximately 68-82% and 67-78% of
annual totals, respectively. During this period, the runoff always exceeds precipitation
which indicates large contributions of glacier/snow-melt water to streamflow. It is
consistent with the existing findings in Tarim River (Yerqiang, Yulongkashi and
Keliya rivers are the major tributaries of Tarim River), which indicated that the melt
water accounted for about half of the annual total streamflow (Fu et al., 2008). The
ET (vegetation cover) in three westerlies-dominated basins are relatively less (scarcer)
than that in other TP basins while the percentages of glacier and seasonal snow cover
are higher in these basins which contribute more melt water to river discharge (Fig.6




and Fig.7). Overall, the SWE in Yerqiang, Yulongkashi and Keliya rivers are higher in
winter than other seasons, but they vary with basins and products which reflect
considerable uncertainties in SWE estimations.

In the Indian monsoon and East Asian monsoon dominated basins, the runoff
concentrates during June-September (or June- October) with precipitation being the
dominant contributor of annual total runoff. For example, the peak values of
precipitation and runoff occur during June-September at Zhimenda station
(contributing about 80% and 74% of the annual totals) while those occur during
June-October at Tangnaihai station (contributing about 78% and 71% of the annual
totals, respectively). The results are quite similar to the related studies in eastern and
southern TP such as Liu (1999), Dong et al. (2007), Zhu et al. (2011), Zhang et al.
(2013), Cuo et al. (2014). The vegetation cover (ET) in most basins is denser (higher)
than that in the westerlies-dominant basins. Moreover, the seasonal snow mainly
covers from mid-autumn to spring and correspondingly the SWE is relatively higher
in these months in all basins except for Yellow River above Xining station, Salwee
River above Jiayuqiao station and Brahmaputra River above Nuxia and Yangcun
stations.

**3.3 Trends of basin-wide water budget components for the TP basins**
The Q, P and $ET_{wb}$ all ascended under regional warming during the past 30 years in
the westerlies-dominated basins (Fig.8), except for P in the Yerqiang River basin
(Kulukelangan station). The aridity index (PET/P), which is an indicator for the
degree of dryness, slightly declined in all basins in northwestern TP. Although both P
and PET were found increase in the Keliya River basin since the 1980s (Shi et al.,



2003; Yao et al., 2014), the declined PET/P is, to some extent, attributed to the
ascending P exceed the increase in PET. The climate moistening (Shi et al., 2003) in
the headwaters of these inland rivers would be beneficial to the water resources and
oasis agro-ecosystems in the middle and lower basins. The increase in streamflow was
also found in most tributaries of the Tarim River (Sun et al., 2006; Fu et al., 2010;
Mamat et al., 2010). Moreover, the westerlies, revealed by the Asian Zonal
Circulation Index ($60^o$-$150^o$ E), slightly enhanced (linear trend: 0.21) over the period
1982-2011 (Fig.9). With the strengthening westerlies, more water vapor may be
transported and fell as precipitation or snow in northwestern TP (e.g., the eastern
Pamir region). Both SWE products (VIC_IGSNRR simulated and GlobaSnow-2
product) showed slightly increase across these basins with rising seasonal snow
covers and glaciers (Yao et al., 2012). More precipitation was transformed into snow
/glacier and the runoff coefficient (Q/P) exhibited decrease with precipitation
obviously increased (Fig.8). In addition, the transpiration in these basins might
decrease with vegetation degradation as revealed by the NDVI and LAI (Yin et al.,
2016) but the atmospheric evaporative demand indicated by CRU PET increased
(significantly increase in the Yulongkashi and Keliya rivers) during the period

435   1982-2011.



In the East Asian monsoon dominated basins, there are two types of change for
basin-wide water budget components. For example, P and Q slightly decreased in the
upper Yellow River (Tangnihai, Huangheyan and Jimai stations) and Yalong River
(Yajiang station) but increased in other basins (Zelingou, Gandatan, Xining, Tongren
and Zhimenda stations) over the period of 1982-2011 (Fig.10). The declind Q and P in





the upper Yellow and Yalong Rivers (locates at the eastern Tibetan Plateau) were
consistent with that found by Cuo et al. (2013, 2014) and Yang et al. (2014), and were
in line with the weakening East Asian Summer Monsoon (linear slope: -0.01) (Fig.9).
The vegetation turned green while $ET_{wb}$ and PET increased in all East Asian
monsoon dominated basins (except for $ET_{wb}$ in the basins above Tongren and Yajing
stations) with the significantly ascending air temperature during the period 1982-2011.
The aridity index (PET/P) decreased in all basins except for the upper Yellow River
basin above Jimai station and the upper Yalong River basin above Yajiang station.
Moreover, both the runoff coefficients and SWE decreased except for the Bayin River
above Zelingou station and the upper Yellow River above Tongren station in the East
Asian monsoon dominated basins.

The P, $ET_{wb}$ and Q also increased in the Indian monsoon-dominated basins (except
for $ET_{wb}$ in the basin above Yangcun station) such as Salween River and
Brahmaputra River (Fig.11), which are in line with the strengthening (linear trend:
0.01) of the Indian summer monsoon (revealed by the Indian Ocean Dipole Mode
Index) during the specific period 1982-2011 (Fig.9). For example, at Jiayuqiao station,
the annual streamflow showed a slightly increasing trend which was consistent with
that examined by Yao et al. (2012) during the period 1980-2000. The vegetation status,
revealed by NDVI and LAI, turned better asscoiated with the ascending air
temperature. The aridity index (PET/P) decreased in all basins except for the
Brahmaputra River above Tangjia station, which indicated that most basins in the
Indian monsoon-dominated regions turned wet over the period of 1982-2011. The
runoff coefficient (Q/P) increased at Gongbujiangda and Nuxia while decreased at
Jiayuqiao, Pangduo, Tangji and Yangcun stations. Moreover, the basin-wide SWE





declined in the upper Salween River and Brahmaputra River above Pangduo, Tangjia
and Gongbujiangda stations while increased in Brahmaputra River above Nuxia and
Yangcun stations.

**3.4 Uncertainties**
The results may unavoidably associate with some uncertainties inherited from the
multi-source datasets used. The primary sources of uncertainty may arise from the
precipitation inputs. We compared the seasonal cycles and annual trends in different
precipitation products, i.e. CMA_ P, IGSNRR_P and TRMM_P (and their
calculated $ET_{wb}$ from the water balance) during the period 2000-2011 (Fig. 12 and
Fig. 13). We found there are some uncertainties among different precipitation
products and thus among their estimated $ET_{wb}$, especially in the westerlies-dominated
basins. However, for each basin, the seasonal cycles of precipitation (and their
calculated  $ET_{wb}$) calculated from different products are overall similar (especially for
the observation-based products, CMA_P and IGSNNR_P). The signs of trend for
annual CMA_P and IGSNRR_P (and their calculated $ET_{wb}$) are consistent in most
river basins (i.e., 14 out 18 basins for two precipitation products and 17 out 18 basins
for their calculated $ET_{wb}$) during the period 1982-2011. The consistency of trends
between two precipitation products, to some extent, revealed that the trends in
CMA_P were not obviously influenced by the changing density of rain gauges in TP
basins. Although some uncertainties exist due to limited and unevenly distributed
meteorological stations used in the plateau and the influences of complex terrain,
CMA_P is still the best observation-based precipitation product nowadays in China
which could be applied to hydrological studies in the TP.




Although the seasonal cycles of $ET_{wb}$ could be captured by GLEAM_E and VIC_E,
they still have considerable uncertainties at some stations (e.g., Numaitilangan,
Gongbujiangda and Nuxia) (Fig.5). Compared to the annual trend of $ET_{wb}$ (Table 4),
most ET products (including the well-performed GLEAM_E and VIC_E) could not
detect the decreasing trends in 7 out of 18 basins (Kulukelangan, Tongguziluoke,
Xining, Tongren, Jimai, Nuxia and Gongbujiangda) due to their different forcing data,
algorithm used as well as varied spatial-temporal resolutions (Xue et al., 2013; Li et
al., 2014; Liu et al., 2016a). In particular, it is well known that land surface models
have some difficulties (e.g., parameter tuning in boundary layer schemes) when
applying to the TP, even though they sometimes have good performances in different
regions/basins (Xia et al., 2012; Bai et al., 2016). For example, Xue et al. (2013)
indicated that GNoah_E underestimated the $ET_{wb}$ in the upper Yellow River and
Yangtze River basins on the Tibetan Plateau mainly due to its negative-biased
precipitation forcing. We thus only used $ET_{wb}$ in the trend detection of water budget
components in Fig.8, Fig.10 and Fig.11 in this study. The two SWE products also
showed large uncertainty with respect to both their seasonal cycles and trends. The
VIC_IGSNRR simulated and GlobaSnow-2 SWEs have not been validated in the TP
due to the lack of snow water equivalent observations, but in some basins (e.g.,
Zelingou and Numaitilangan) they showed similar seasonal cycles and annual trends.

The interpolation of missing values of runoff with VIC_IGSNRR simulated runoff
and the gridded precipitation data (which interpolated from limited gauged
precipitation over the plateau) also introduced uncertainties. There are also
considerable uncertainties arising from empirical extending the ET series back prior



to the GRACE era. However, the trends in $ET_{wb}$ have not significantly affected by
erroneous trends in the precipitation inputs to the bias-correction based water balance
calculation. For example, the trends in CMA_P and IGSNRR_P are opposite in few
basins (No. 01, 07, 08, 13 in Fig. 13), but the trends in their calculated $ET_{wb}$ are both
consistent for each basin. It is, to some extent, certified the effectiveness of the bias
correction-based ET-estimate approach. With these caveats, we can interpret the
general hydrological regimes and their responses to the changing climate in the TP
basins from solely the perspective of multi-source datasets, which are comparable to
the existing studies based on the in situ observations and complex hydrological
modeling.

**4    Summary**
In this study, we investigated the seasonal cycles and trends of water budget
components in 18 TP basins during the period 1982-2011, which is not well
understood so far due to the lack of adequate observations in the harsh environment,
through integrating the multi-source global/regional datasets such as gauge data,
satellite remote sensing and land surface model simulations. By using a two-step bias
correction procedure, we calculated the annual basin-wide $ET_{wb}$ through the water
balance approach considering the impacts of glacier and water storage change. We
found that the GLEAM_E and VIC_E perform better relative to other products against
the calculated $ET_{wb}$.

From the Budyko framework perspective, the general water and energy budgets are
different in the westerlies-dominated (with higher aridity index, runoff coefficient and
glacier cover), the Indian monsoon-dominated and the East Asian





monsoon-dominated (with higher air temperature, vegetation cover and
evapotranspiration) basins. In the 18 TP basins, precipitation is the major contributor
to the river runoff, which concentrates mainly during June-October (June-August for
the westerlies-dominated basins, June-September or June to October for the Indian
monsoon-dominated and the East Asian monsoon-dominated basins). The basin-wide
SWE is relatively high from mid-autumn to spring for all 18 TP basins except for
Keliya River and Brahmaputra River above the Nuxia and Yangcun stations. The
vegetation cover is relatively less whereas snow/glacier cover is more in the
westerlies-dominant basins compared to other basins. During the period 1982-2011,
we found that the P, Q and $ET_{wb}$ increased across most of the basins in Tibetan
Plateau; receded at some tributaries located at the upper Yellow River and Yalong
River due to the weakening East Asian monsoon. The aridity index (PET/P) exhibited
decrease in most TP basins which corresponded to the warming and moistening
climate in the TP and western China. Moreover, the runoff coefficient (Q/P) declined
in most basins which may be, to some extent, due to ET increase induced by
vegetation greening and the influences of snow and glacier changes. Although there
are considerable uncertainties inherited from multi-source data used, the general
hydrological regimes in the TP basins could be revealed, which are consistent to the
existing results obtained from in situ observations and complex land surface modeling.
It indicated the usefulness of integrating the multiple datasets (e.g., in situ
observations, remote sensing-based products, reanalysis outputs, land surface model
simulations and climate model outputs) for hydrological applications. The
generalization here could be helpful for understanding the hydrological cycle and
supporting sustainable water resources management and eco-environment protection
in the Tibetan Plateau under global warming.






*Author contributions*. Wenbin Liu and Fubao Sun developed the idea to see the
general water budgets in the TP basins from the perspective of multisource datasets.
Wenbin Liu collected and processed the multiple datasets with the help of Yanzhong
Li, Guoqing Zhang, Wee Ho Lim, Hong Wang as well as Peng Bai, and prepared the
manuscript. The results were extensively commented and discussed by Fubao Sun,
Jiahong Liu and Yan-Fang Sang.

*Acknowledgements*. This study was supported by the National Key Research and
Development Program of China (2016YFC0401401 and 2016YFA0602402),National
Natural Science Foundation of China (41401037 and 41330529), the Open Research
Fund of State Key Laboratory of Desert and Oasis Ecology in Xinjiang Institute of
Ecology and Geography, Chinese Academy of Sciences (CAS), the CAS Pioneer
Hundred Talents Program (Fubao Sun), the CAS President's International Fellowship
Initiative (2017PC0068) and the program for the "Bingwei" Excellent Talents from
the Institute of Geographic Sciences and Natural Resources Research, CAS. We are
grateful to the NASA MEaSUREs Program (Sean Swenson) for providing the
GRACE land data processing algorithm. The basin-wide water budget series in the TP
Rivers used in this study are available from the authors upon request
([liuwb@igsnrr.ac.cn](mailto:liuwb@igsnrr.ac.cn)). We thank the editors and reviewers for their invaluable
comments and constructive suggestions.

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



**Table 1**: Overview of multi-source datasets applied in this study

| Data category | Data source | Spatial resolution | Temporal resolution | Available period used | Reference |
|---|---|---|---|---|---|
| Runoff (Q) | Observed, National Hydrology Almanac of China | — | Daily | 1982–2011 | — |
| | VIC_IGSNRR simulated | 0.25° | Daily | 1982–2011 | Zhang et al. (2014) |
| Precipitation (P) | Observed, CMA | 0.5° | Monthly | 1982–2011 | — |
| | TRMM 3B43 V7 | 0.25° | Monthly | 2000–2011 | Huffman et al. (2012) |
| | IGSNRR forcing | 0.25° | Daily | 1982–2011 | Zhang et al. (2014) |
| Temperature (Temp.) | Observed, CMA | 0.5° | Monthly | 2000–2011 | — |
| Terrestrial storage change | GRACE-CSR | Approx.300-400 km | Monthly | 2002–2011 | Tapley et al. (2004) |
| (ΔS) | GRACE-GFZ | Approx.300-400 km | Monthly | 2002–2011 | Tapley et al. (2004) |
| | GRACE-JPL | Approx.300-400 km | Monthly | 2002–2011 | Tapley et al. (2004) |
| Potential evaporation (PET) | CRU | 0.5° | Monthly | 1982–2011 | Harris et al. (2013) |
| Actual evaporation (ET) | MTE_E | 0.5° | Monthly | 1982–2011 | Jung et al. (2010) |
| | VIC_E | 0.25° | Daily | 1982–2011 | Zhang et al. (2014) |
| | GLEAM_E | 0.25° | Daily | 1982–2011 | Miralles et al. (2011) |
| | PML_E | 0.5° | Monthly | 1982–2011 | Zhang Y et al. (2016) |
| | Zhang_E | 8 km | Monthly | 1983–2006 | Zhang et al. (2010) |
| | GNoah_E | 1.0° | 3 hourly | 1982–2011 | Rui (2011) |
| NDVI | GIMMS NDVI dataset | 8 km | 15 daily | 1982–2011 | Tucker et al. (2005) |
| LAI | GLASS LAI Product | 0.05° | 8 daily | 1982–2011 | Liang and Xiao (2012) |
| Snow Cover | TP Snow composite Products | 500 m | Daily | 2005-2013 | Zhang et al. (2012) |
| SWE | VIC_IGSNRR simulated | 0.25° | Daily | 1982–2011 | Zhang et al. (2014) |
| | GlobSnow-2 Product | 25 km | Daily | 1982–2011 | Takala et al. (2011) |
| Glacier Area | the Second Glacier Inventory Dataset of China | — | — | 2005 | Guo et al. (2014) |



**Table 2:** Main features of the 18 TP river basins used in this study. The precipitation and temperature statistics for each basin were calculated from the observed CMA datasets while the NDVI and LAI statistics were extracted from the GIMMS NDVI dataset and GLASS LAI product. The GA% and SC% represented the percentages of multiyear-mean glacier cover and snow cover in each basin which were calculated from the Second Glacier Inventory Dataset of China and the daily TP snow cover dataset (2005-2013)

| No. | Station | Altitude (m) | River name | Drainage area (km²) | Multiyear-mean (1982-2011) and basin-averaged parameters | | | | | | |
|---|---|---|---|---|---|---|---|---|---|---|---|
| | | | | | Q (mm/yr) | Prec. (mm/yr) | Temp.(°C/yr) | NDVI | LAI | GA% | SC% |
| 01 | Kulukelangan | 2000 | Yerqiang | 32880.00 | 158.60 | 128.34 | -5.68 | 0.05 | 0.03 | 10.97 | 35.03 |
| 02 | Tongguziluoke | 1650 | Yulongkashi | 14575.00 | 151.56 | 134.04 | -4.07 | 0.06 | 0.04 | 23.27 | 35.95 |
| 03 | Numaitilangan | 1880 | Keliya | 7358.00 | 103.18 | 137.14 | -4.78 | 0.06 | 0.03 | 10.86 | 29.16 |
| 04 | Zelingou | 4282 | Bayin | 5544.00 | 41.42 | 340.68 | -4.98 | 0.13 | 0.09 | 0.09 | 21.22 |
| 05 | Gadatan | 3823 | Yellow | 7893.00 | 200.95 | 566.01 | -4.60 | 0.34 | 0.54 | 0.13 | 14.94 |
| 06 | Xining | 3225 | Yellow | 9022.00 | 99.90 | 503.74 | 0.97 | 0.36 | 0.70 | 0.00 | 10.06 |
| 07 | Tongren | 3697 | Yellow | 2832.00 | 149.36 | 533.25 | -1.37 | 0.39 | 0.83 | 0.00 | 9.42 |
| 08 | Tainaihai | 2632 | Yellow | 121972.00 | 159.48 | 540.32 | -2.40 | 0.34 | 0.72 | 0.09 | 15.89 |
| 09 | Huangheyan | 4491 | Yellow | 20930.00 | 31.18 | 386.42 | -4.81 | 0.23 | 0.61 | 0.00 | 17.25 |
| 10 | Jimai | 4450 | Yellow | 45015.00 | 85.50 | 441.48 | -4.16 | 0.26 | 0.52 | 0.00 | 20.05 |
| 11 | Yajiang | 2599 | Yalong | 67514.00 | 237.66 | 717.05 | -0.23 | 0.43 | 0.80 | 0.15 | 18.36 |
| 12 | Zhimenda | 3540 | Yangtze | 137704.00 | 96.23 | 405.66 | -4.83 | 0.20 | 0.26 | 0.96 | 17.87 |
| 13 | Jiaoyuqiao | 3000 | Salween | 72844.00 | 364.26 | 620.88 | -1.89 | 0.29 | 0.44 | 2.02 | 23.73 |
| 14 | Pangduo | 5015 | Brahmaputra | 16459.00 | 348.31 | 544.59 | -1.53 | 0.27 | 0.33 | 1.66 | 23.33 |
| 15 | Tangjia | 4982 | Brahmaputra | 20143.00 | 350.61 | 555.17 | -1.89 | 0.27 | 0.34 | 1.39 | 21.83 |
| 16 | Gongbujiangda | 4927 | Brahmaputra | 6417.00 | 586.96 | 692.06 | -4.24 | 0.27 | 0.36 | 4.12 | 25.99 |
| 17 | Nuxia | 2910 | Brahmaputra | 191235.00 | 307.38 | 401.35 | -0.73 | 0.22 | 0.25 | 1.90 | 13.50 |
| 18 | Yangcun | 3600 | Brahmaputra | 152701.00 | 163.25 | 349.91 | -0.87 | 0.19 | 0.18 | 1.28 | 10.52 |






**Table 3**: Annual-averaged water storage changes (ΔS) in 18 TP basins derived from GRACE retrievals (2002–2013) from three different processing centers (CSR, GFZ and JPL)

| Basin | Water storage Change (ΔS,mm) | | |
|---|---|---|---|
| | CSR | GFZ | JPL |
| Kulukelangan | -0.16 | -0.16 | -0.00 |
| Tongguziluoke | 0.10 | 0.10 | 0.28 |
| Numaitilangan | 0.24 | 0.22 | 0.41 |
| Zelingou | 0.63 | 0.41 | 0.69 |
| Gadatan | 0.02 | -0.24 | -0.03 |
| Xining | -0.08 | -0.35 | -0.14 |
| Tongren | -0.13 | -0.41 | -0.21 |
| Tainaihai | 0.12 | -0.16 | 0.10 |
| Huangheyan | 0.60 | 0.35 | 0.70 |
| Jimai | 0.41 | 0.15 | 0.48 |
| Yajiang | -0.23 | -0.50 | -0.21 |
| Zhimenda | 0.57 | 0.38 | 0.78 |
| Jiaoyuqiao | -1.00 | -1.13 | -0.79 |
| Nuxia | -1.42 | -1.44 | -1.31 |
| Pangduo | -1.21 | -1.29 | -1.02 |
| Tangjia | -1.40 | -1.46 | -1.24 |
| Gongbujiangda | -1.61 | -1.67 | -1.47 |
| Yangcun | -1.33 | -1.34 | -1.21 |







**Table 4:** Nonparametric trends for different ET estimates during the period 1982–2006 detected by modified Mann-Kendall test, the bold number showed the detected trend is statistically significant at the 0.05 level

| Basin | ET$_{wb}$ | GLEAM_E | VIC_E | Zhang_E | PML_E | MET_E | GNoah_E |
|---|---|---|---|---|---|---|---|
| Kulukelangan | **-0.09** | 0.09 | **0.18** | – | 0.03 | -0.01 | 0.07 |
| Tongguziluoke | -0.02 | 0.10 | **0.13** | – | 0.03 | **-0.08** | 0.19 |
| Numaitilangan | 0.04 | **0.10** | 0.14 | – | 0.14 | **-0.10** | 0.22 |
| Zelingou | **0.13** | **0.23** | 0.11 | **0.09** | 0.04 | **0.06** | 0.02 |
| Gadatan | -0.09 | 0.25 | 0.070 | -0.10 | -0.01 | **0.06** | -0.07 |
| Xining | -0.06 | **0.54** | 0.01 | -0.08 | 0.01 | 0.02 | -0.06 |
| Tongren | -0.06 | **0.34** | -0.15 | **-0.17** | 0.07 | 0.02 | 0.13 |
| Tainaihai | 0.06 | **0.28** | -0.03 | **-0.11** | 0.04 | **0.05** | 0.04 |
| Huangheyan | 0.08 | **0.19** | -0.01 | **-0.10** | **0.08** | **0.05** | **0.10** |
| Jimai | -0.07 | **0.23** | -0.01 | -0.08 | 0.03 | **0.05** | 0.10 |
| Yajiang | 0.17 | **0.26** | **0.06** | **-0.21** | -0.01 | 0.03 | -0.02 |
| Zhimenda | 0.11 | **0.28** | 0.10 | 0.01 | 0.07 | **0.04** | 0.07 |
| Jiaoyuqiao | **0.18** | **0.28** | 0.10 | **-0.11** | 0.05 | **0.05** | 0.07 |
| Nuxia | **-0.09** | **0.25** | 0.09 | **-0.10** | **0.12** | **0.04** | 0.10 |
| Pangduo | 0.05 | **0.28** | **0.17** | **-0.07** | 0.07 | **0.07** | **0.11** |
| Tangjia | 0.09 | **0.26** | **0.17** | **-0.09** | **0.20** | **0.06** | **0.12** |
| Gongbujiangda | -0.26 | 0.12 | 0.13 | **-0.16** | **0.19** | 0.01 | **0.15** |
| Yangcun | 0.03 | **0.28** | 0.08 | **-0.06** | 0.10 | 0.04 | 0.09 |



**Figure captions:**

**Figure1.** Map of river basins and hydrological gauging stations (green dots) over the

Tibetan Plateau (TP) used in this study. The grey shading shows the topography of TP

in meters above the sea level and the blue shading exhibits the glaciers distribution in

TP extracted from the Second Glacier Inventory Dataset of China.

**Figure 2.** Comparison of VIC_IGSNRR simulated and observed monthly runoff for

Tangnaihai and Panduo stations (a and b) as well as (c) basin-averaged monthly

TRMM, CMA gridded and IGSNRR forcing precipitations for the smallest basin

(Tongren station) over the period 1982-2011. (d) shows the comparison of TRMM

(blue) and IGSNRR forcing (red) precipitations against CMA gridded precipitation for

18 river basins over TP during the period 2000-2011.

**Figure 3.** Comparison of different ET products against the calculated ET through the

water balance method ($ET_{wb}$) at the monthly time scale for 18 TP basins during the

period 1983-2006. The boxplot of monthly estimates of different ET products for 18

TP basins are shown in (a) while the correlation coefficients and

root-mean-square-errors (RMSEs, mm/month) for each ET product relatively to $ET_{wb}$

are exhibited in (b).

**Figure 4**. General water and energy status (a. the perspective of Budyko framework)

and their relationships with glacier (b) and vegetation (c and d) for eighteen TP river

basins (1983-2006). The ET used in this figure is calculated from the bias-corrected

water balance method.

**Figure 5**. Seasonal cycles (1982-2011) of water budget components in westerlies-

dominated (column 1), East Asian monsoon-dominated (columns 2-4) and Indian

monsoon-dominated (columns 5-6) TP basins.

**Figure 6**. Seasonal cycles (1982-2011) of air temperature and vegetation parameters

in westerlies-dominated (column 1), East Asian monsoon-dominated (columns 2-4)

and Indian monsoon-dominated (columns 5-6) TP basins.

**Figure 7**. Seasonal cycles (1982-2011) of snow cover and snow water equivalent

(SWE) in westerlies-dominated (column 1), East Asian monsoon-dominated (columns





2-4) and Indian monsoon-dominated (columns 5-6) TP basins. The snow cover was
extracted from cloud free snow composite product during the period 2005-2013. It
should also be noted that the GlobSnow data are not available for some basins.
**Figure 8**. Sen's slopes of water budget components and vegetation parameters in
westerlies-dominated TP basins during the period of 1982-2011. To clearly exhibit the
nonparametric trends of all variables in one panel, the Sen's Slopes of Q, P, $ET_{wb}$ and
PET have been multiplied by 1/12 (unit: mm/month). The double red stars showed
that the trend was statistically significant at the 0.05 level.
**Figure 9**. Linear and non-parametric trends of westerly, Indian monsoon and East
Asian summer monsoon during the period 1982-2011 revealed prospectively by the
Asian Zonal Circulation Index, Indian Ocean Dipole Mode Index and East Asian
Summer Monsoon Index.
**Figure 10**. Similar to Figure 8 but for East Asian monsoon-dominated TP basins. It
should be noted that the GlobSnow data are not available for some basins. The double
red stars showed that the trend was statistically significant at the 0.05 level.
**Figure 11**. Similar to Figure 8 but for Indian monsoon-dominated TP basins. It should
be noted that the GlobSnow data are not available for some basins. The double red
stars showed that the trend was statistically significant at the 0.05 level.
**Figure 12**. Uncertainties in seasonal cycles of ETwb calculated from three precipitation
products (CMA gridded, IGSNRR_Forcing and TRMM precipitation) in 18 TP basins.
The comparisons were conducted during the period 2000-2011 when TRMM data was
available.
**Figure 13**. Uncertainties in annual trends of $ET_{wb}$ (b) calculated from two precipitation
products (CMA gridded and IGSNRR_Forcing) (a) in 18 TP basins. The comparisons
were conducted during the period 1982-2011(TRMM data was not available for the
whole period).





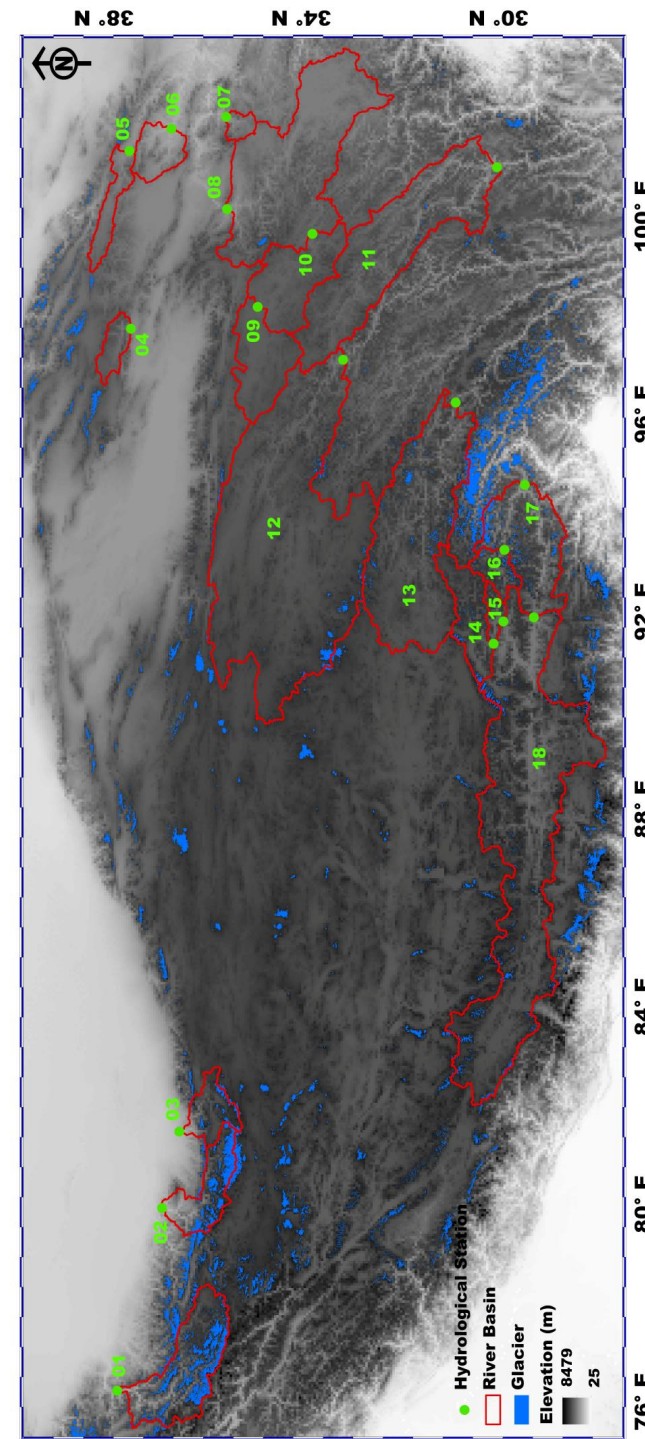

**Figure 1.** Map of river basins and hydrological gauging stations (green dots) over the Tibetan Plateau (TP) used in this study. The grey shading shows the topography of TP in meters above the sea level and the blue shading exhibits the glaciers distribution in TP extracted from the Second Glacier Inventory Dataset of China.







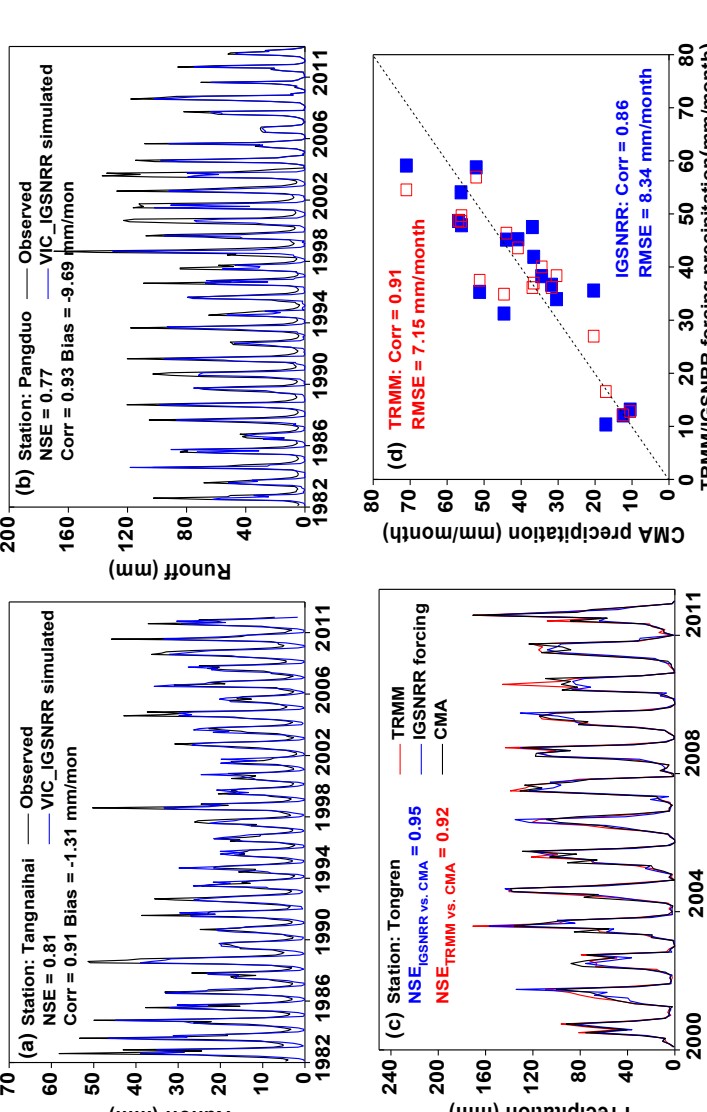

**Figure 2.** Comparison of VIC_IGSNRR simulated and observed monthly runoff for Tangnaihai and Panduo stations (a and b) as well as (c) basin-averaged monthly TRMM, CMA gridded and IGSNRR forcing precipitations for the smallest basin (Tongren station) over the period 1982-2011. (d) shows the comparison of TRMM (blue) and IGSNRR forcing (red) precipitations against CMA gridded precipitation for 18 river basins over TP during the period 2000-2011.





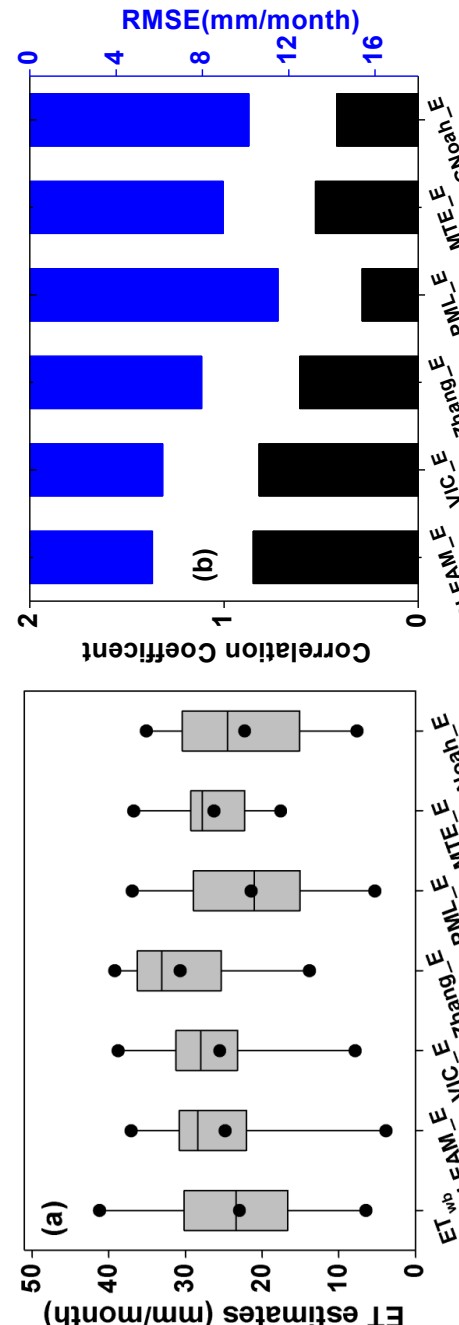

**Figure 3.** Comparison of different ET products against the calculated ET through the water balance ($ET_{wb}$) at the monthly time scale for 18 river basins over the Tibetan Plateau during the period 1983–2006. The boxplot of monthly estimates of different ET products for 18 TP basins are shown in (a) while the correlation coefficients and root-mean-square-errors (RMSEs, mm/month) for each ET product relatively to $ET_{wb}$ are exhibited in (b).

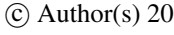


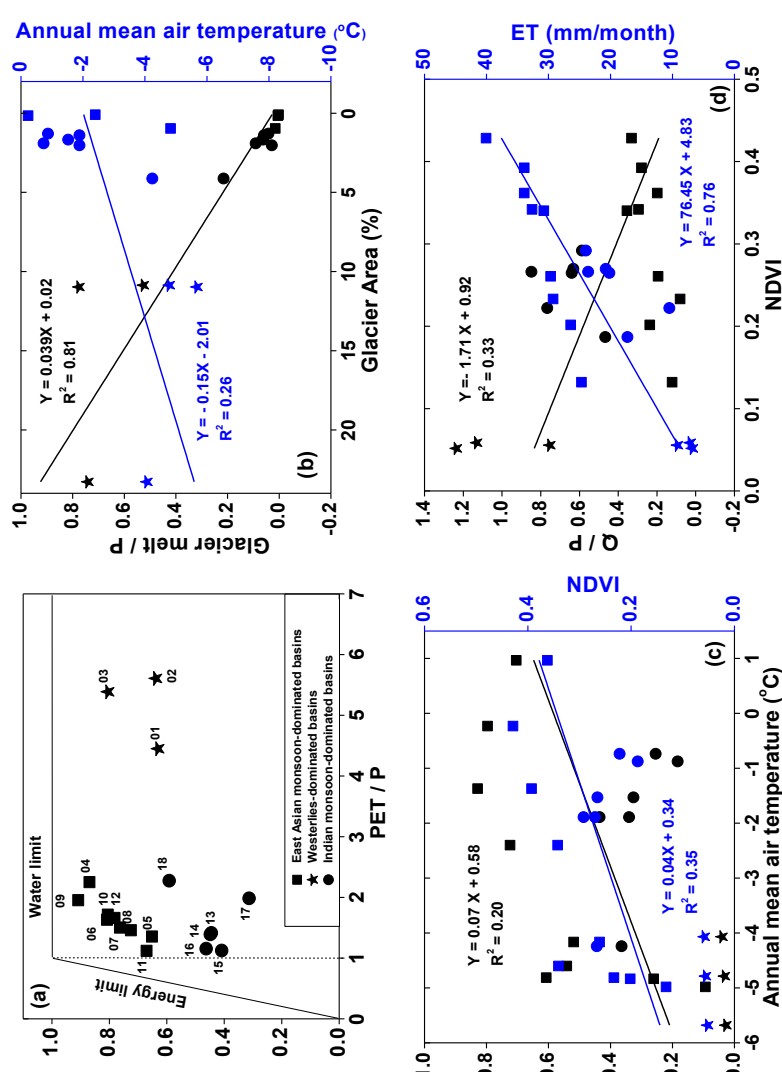

**Figure 4.** General water and energy status (a. the perspective of Budyko framework) and their relationships with glacier (b) and vegetation (c and d) for eighteen TP river basins (1983-2006). The ET used in this figure is calculated from the bias-corrected water balance method.





**Figure 5.** Seasonal cycles (1982–2011) of water budget components in westerlies-dominated (column 1), East Asian monsoon-dominated (columns 2–4) and Indian monsoon-dominated (columns 5–6) TP basins.





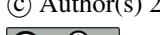



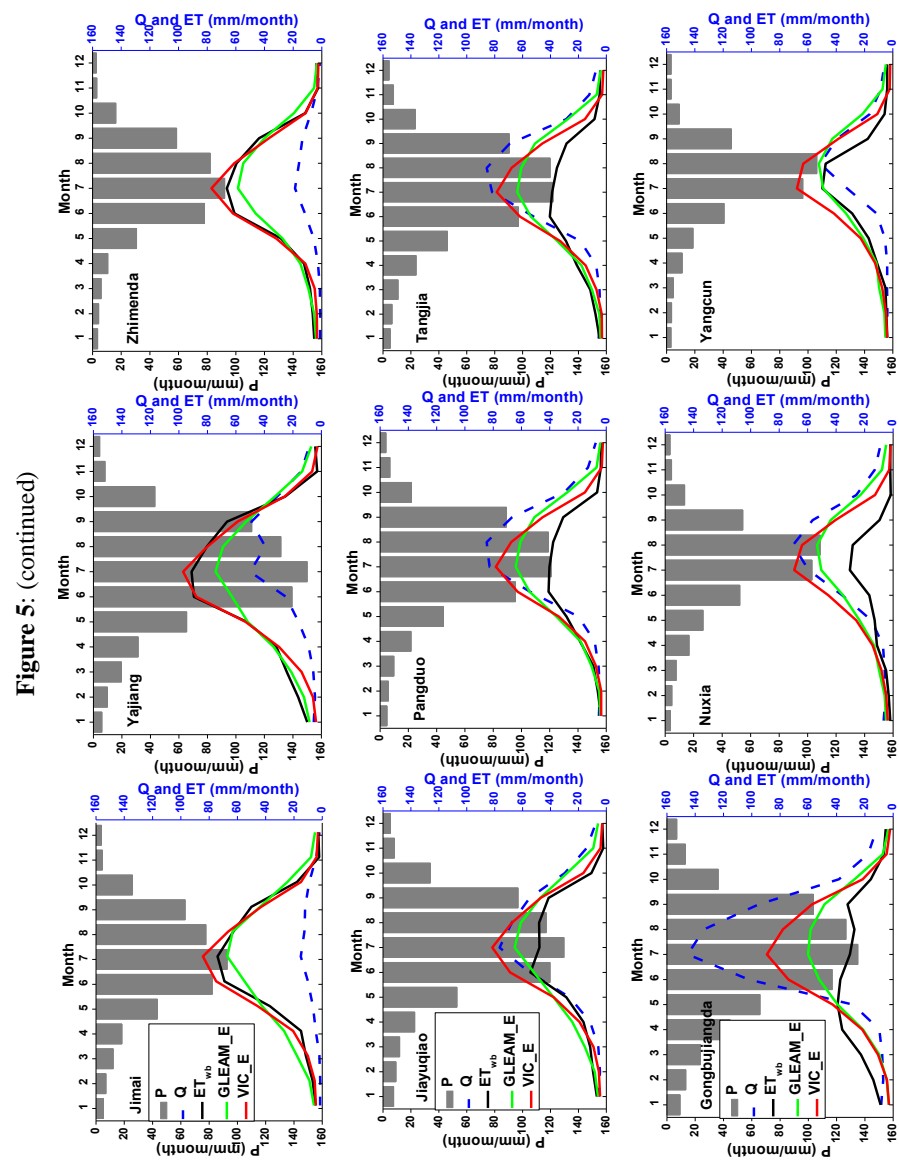

**Figure 5:** (continued)





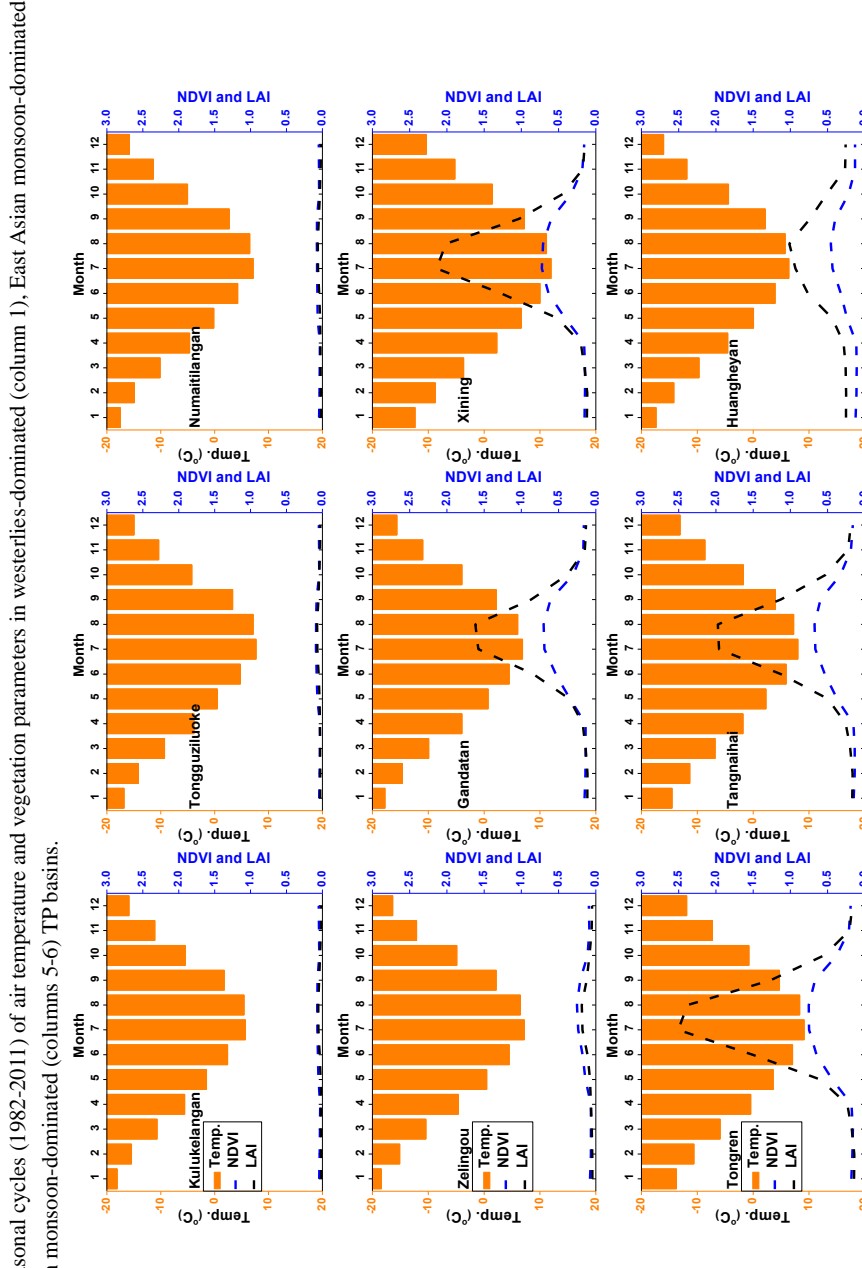

**Figure 6.** Seasonal cycles (1982–2011) of air temperature and vegetation parameters in westerlies-dominated (column 1), East Asian monsoon-dominated (columns 2–4) and Indian monsoon-dominated (columns 5–6) TP basins.



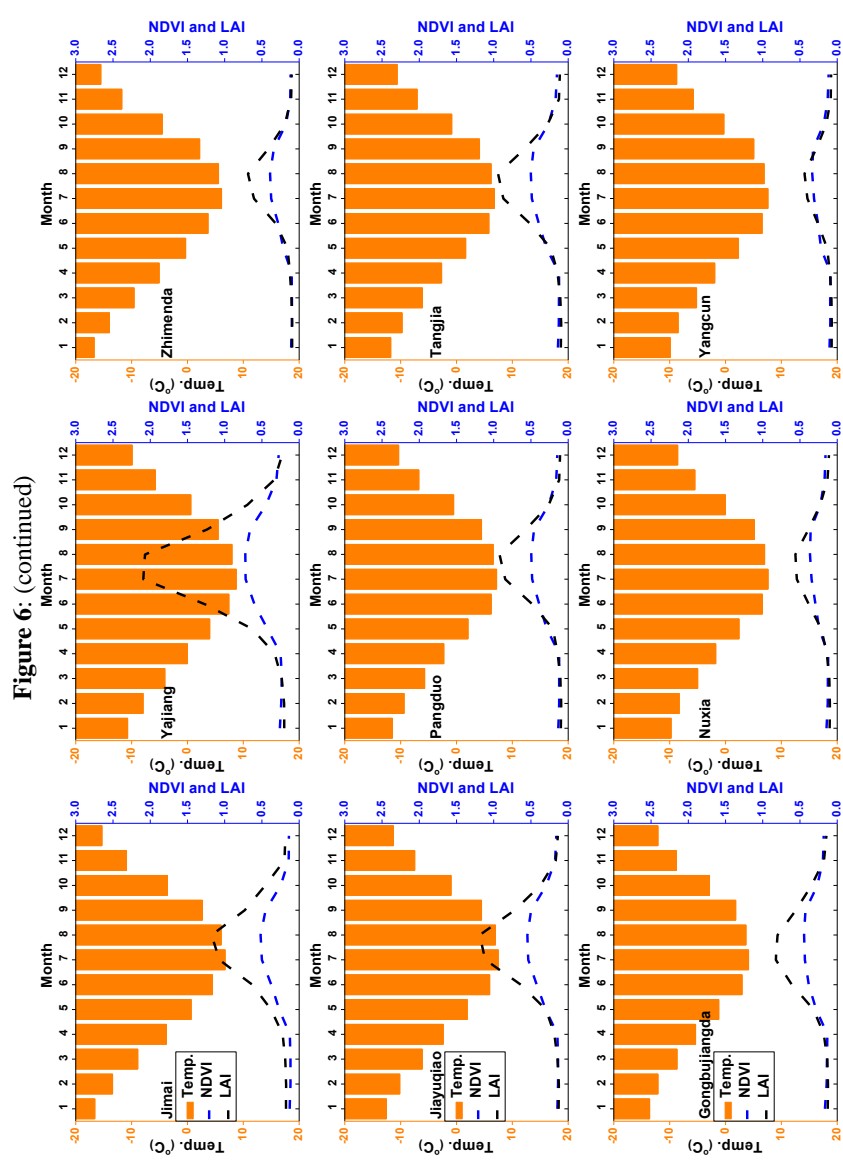

**Figure 6:** (continued)



**Figure 7**. Seasonal cycles (1982–2011) of snow cover and snow water equivalent (SWE) in westerlies-dominated (column 1), East Asian monsoon- dominated (columns 2-4) and Indian monsoon-dominated (columns 5-6) TP basins. The snow cover was extracted from cloud free snow composite product during the period 2005-2013. It should also be noted that the GlobSnow data are not available for some basins.




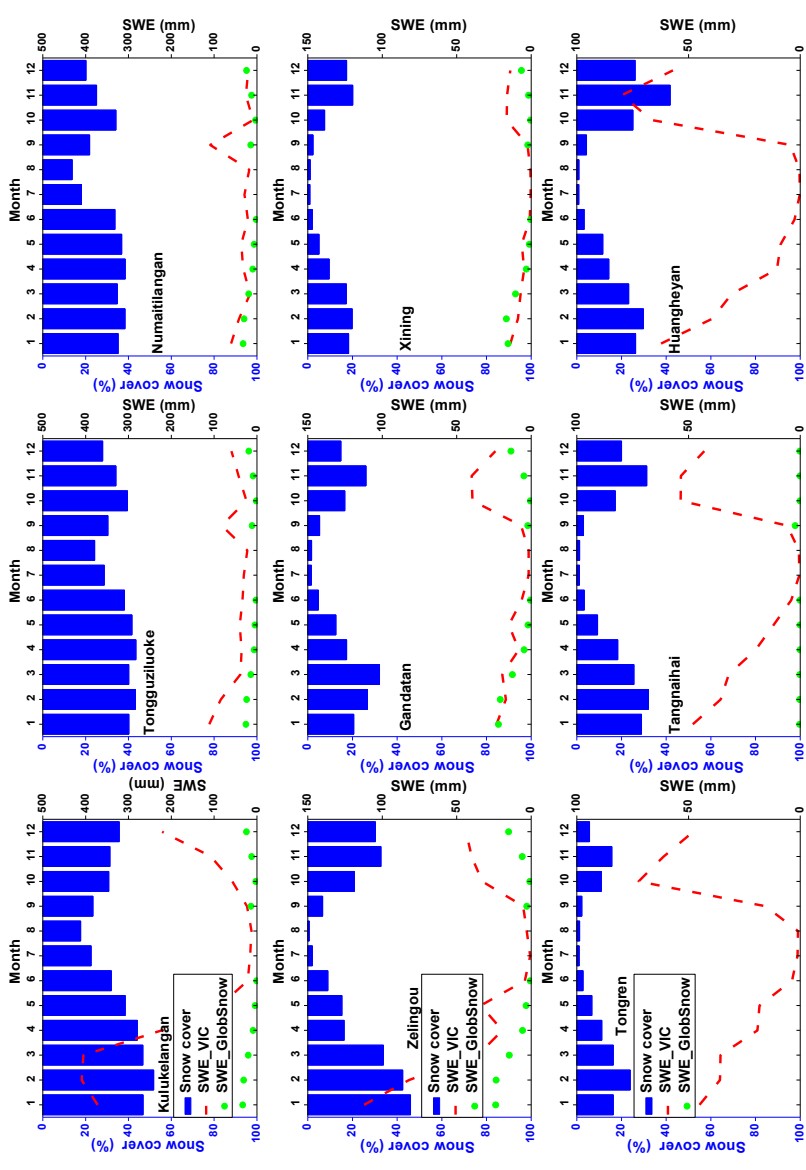

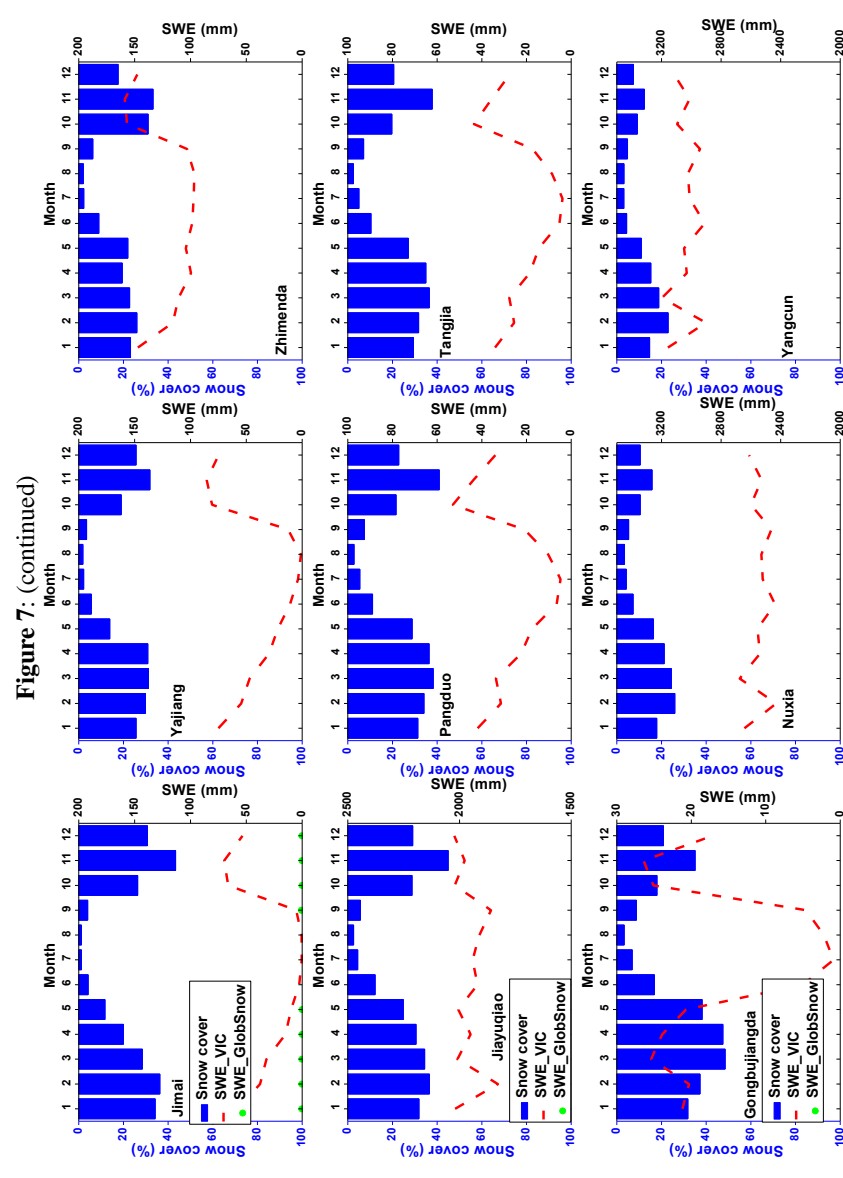

**Figure 7**: (continued)










**Figure 8.** Sen's slopes of water budget components and vegetation parameters in westerlies-dominated TP basins during the period of 1982-2011. To clearly exhibit the nonparametric trends of all variables in one panel, the Sen's Slopes of Q, P, $ET_{wb}$ and PET have been multiplied by 1/12 (unit: mm/month). The double red stars showed that the trend was statistically significant at the 0.05 level.





**Figure 9**. Linear and non-parametric trends of westerly, Indian monsoon and East Asian summer
monsoon during the period 1982-2011 revealed prospectively by the Asian Zonal Circulation
Index, Indian Ocean Dipole Mode Index and East Asian Summer Monsoon Index.






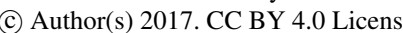

**Figure 10.** Similar to Figure 8 but for East Asian monsoon-dominated TP basins. It should be noted that the GlobSnow data are not available for some basins.





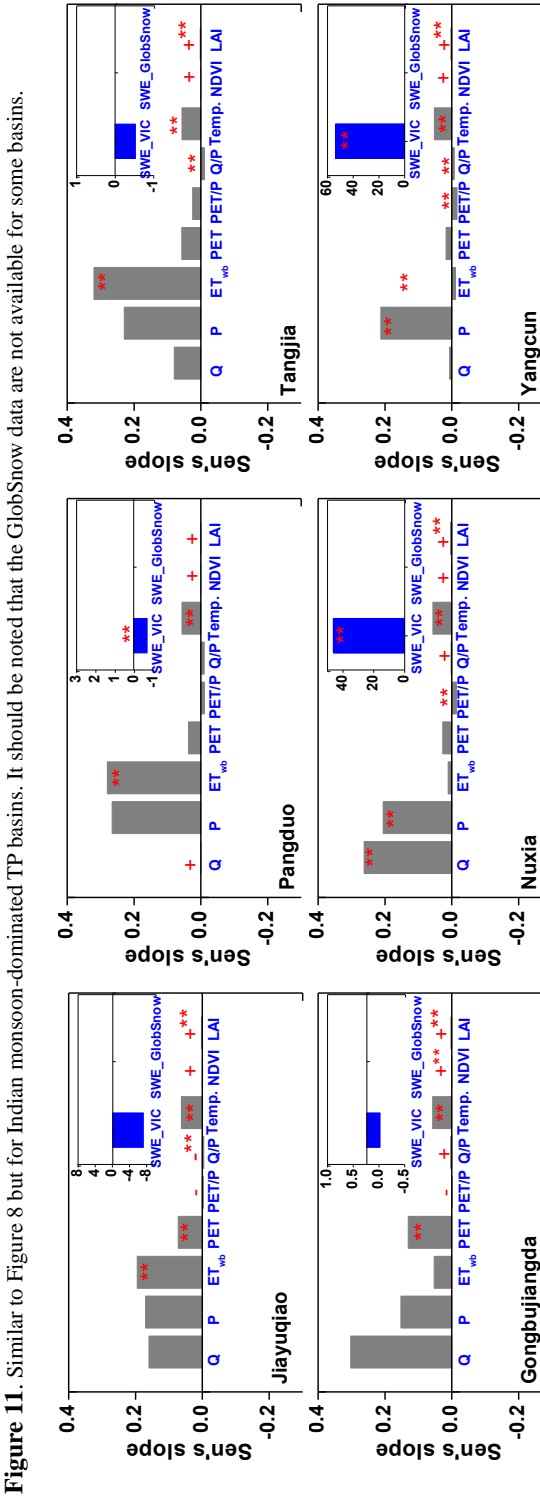

**Figure 11.** Similar to Figure 8 but for Indian monsoon-dominated TP basins. It should be noted that the GlobSnow data are not available for some basins.





**Figure 12.** Uncertainties in seasonal cycles of ET$_{wb}$ calculated from three precipitation products (CMA gridded, IGSNRR_Forcing and TRMM precipitation) in 18 TP basins. The comparisons were conducted during the period 2000-2011 when TRMM data was available.



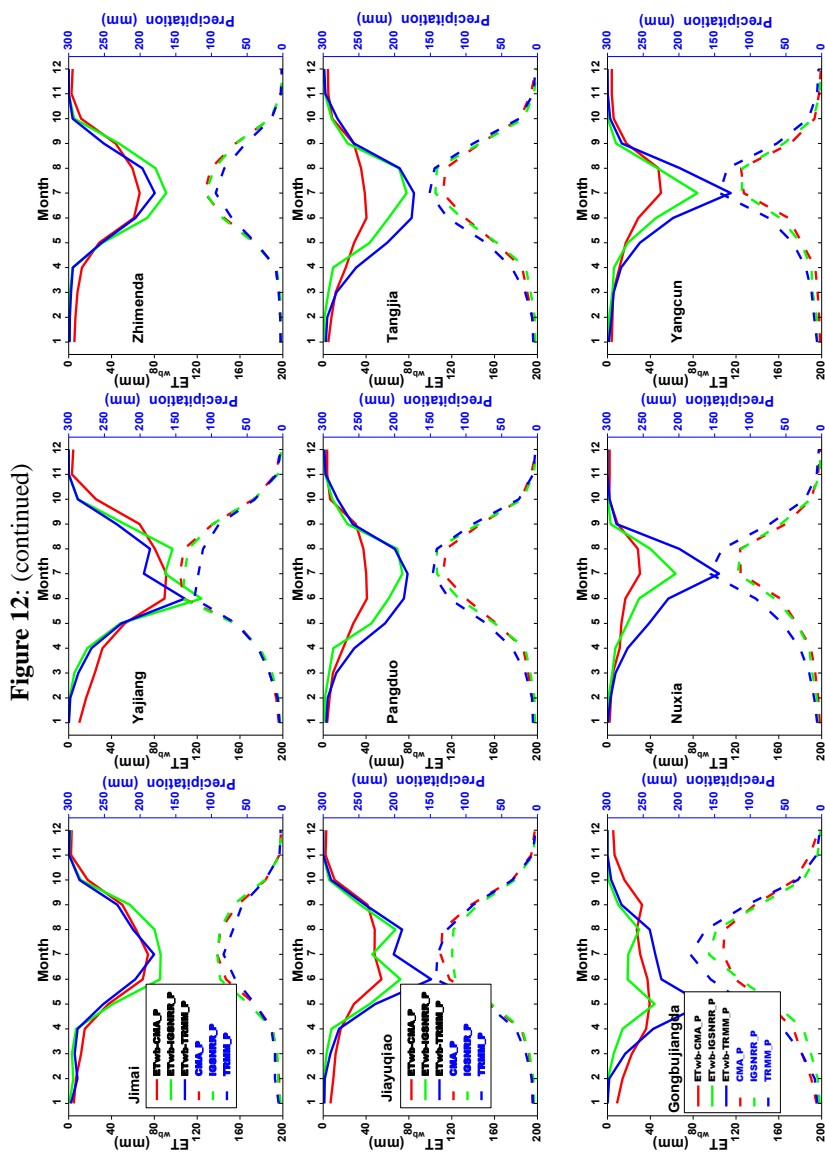

**Figure 12:** (continued)





**Figure 13.** Uncertainties in annual trends of $ET_{wb}$ (b) calculated from two precipitation products (CMA gridded and IGSNRR_Forcing) (a) in 18 TP basins. The comparisons were conducted during the period 1982-2011(TRMM data was not available for the whole period).