# Peer review of "Investigating basin-scale water budget dynamics in 18 rivers across"

_Hydrology and Earth System Sciences, 2017_

## Referee Comment (RC1) · Anonymous Referee #1 · 28 Aug 2017

General comments: This manuscript investigates the seasonal cycles and trends of water budgets over 18 river basins in the Tibetan Plateau using a wide range of datasets from satellite-observed, land-surface-models simulated, reanalysis and upscaled results of in-situ observations. Prior to seeking the general hydrological features over 18 basins under the Budyko framework, they first assessed the accuracy of six ET products using the water-balance-method derived ET values. They also found that P, Q and ET generally increased in past 30 years in most basins, demonstrating an overall moistening trend in TP. While a quantitative illustrating the uncertainties in the results is very difficult due to the lack of in-situ observation over this remote area, the authors indeed documented the possible uncertainties in the selected datasets, providing a helpful clue for future research. In my view, this MS describes interesting results, which contributes to advancing our understanding of the hydrological cycle regime over such a hydrometrologically important but sparsely-instrumental area. Overall, the MS is nicely structured and presented, and is of a topic that should be of interest to the readers of HESS. I think it is publishable after addressing the comments below. My recommendation is minor revision.

Major comments: I think some analyses over the westly-controlled basins need to be revised because of its special climate patterns compared to other basins. L356-357: "more snow melt contributions" may be due to its special seasonality of precipitation in the westlies-controlled basins rather than its "colder" status. By the way, could you show the annual mean temperature for each basin in Figure 6 to support they are indeed "colder"? Also, I think the (limited) water availability plays a more important role than the heat stress (i.e. colder status) in leading to a relatively less vegetation over such basins. From Figure 4 c and d, it appears than $R2$ between ET and NDVI (0.76) is much higher than that between T and NDVI (0.35).

Specific comments: L3: "river basins" may be more appropriate than "rivers" L67: Most stations are located only in eastern TP and few of them situated in the western part. It would be better if you can highlight such a challenge. L71: It seems that "snow depth" is recorded by these stations. I suggest deleting such a term. L77: labor and/or technical support for maintaining in-situ observation is also a great challenge in addition to the high cost. L419: reword "attributed to the ascending P exceed the increase in PET" as "due to the higher rates of the increase of P than that of PET" L426: change "precipitation" to "rain"? L463: The increase of PET/P may be consistent to the changes in moisture flux of TP, as illustrated by Gao et al. (2014). L553: change "; receded at some tributaries" to "with the exception of some tributaries of" L555: revise "a decrease trend" and change "corresponded" to "corresponds" L562: change "indicated" to "indicates" L567: delete "under global warming"

Reference: Gao. Y., Cuo, L., Zhang, Y. 2014. Changes in Moisture Flux over

the Tibetan Plateau during 1979–2011 and Possible Mechanisms. J. Clim., doi: 10.1175/JCLI-D-13-00321.1

---

## Referee Comment (RC2) · Anonymous Referee #2 · 15 Sep 2017

General comments: accepted after minor revision. This manuscript investigated the dynamics of water budgets of the 18 river basins over Tibetan Plateau (TP) by multi-source datasets including in situ observations, satellite retrievals, reanalysis outputs and land surface model outputs. The actual evapotranspiration was estimated using a water balance-based two-step procedure which considered the changes in basin-scale water storage at the annual scale. Their results show that precipitation is the major contributor to the runoff in TP basins and the weakening East Asian Monsoon mainly affected the increased water budget components. It offers a helpful insight towards understanding the water and energy budgets and sustainability of water resource management practices in the data-sparse TP region based on the current-existing multisource datasets. Overall, the topic and results of the manuscript are very interesting and meaningful, and fit well with the scope of HESS. It is also well-written and organized. I have also noticed that the manuscript is a resubmission. After reading the review comments and the corresponding responses in the last time, I found the manuscript has been significantly improved. In my opinion, the manuscript could be considered for publication after some minor revisions this time.

Specific comments: Some typo errors given below need to be further corrected. For example, 1. Lines 226-228: The sentence "It has been demonstrated cannot be neglected . . .." should be re-written.

2. Lines 294-295, 306, 307 and 307: The unit of statistical indicators should be uniformed, such as "RMSE=8.34 mm/month" in line 294, "RMSE=5.69 mm month-1". The authors should change them in the whole manuscript.

3. Line 343-352, these sentences should be rewritten to make them more readable.

4. Lines 346, the use of "∼" and "—" should also be unified for the entire manuscript.

5. Line 350, Table 1 or Table 2?

6. Lines 420: "change" should be "changes".

7. Line 562: "indicates"??

Please also note the supplement to this comment:
https://www.hydrol-earth-syst-sci-discuss.net/hess-2017-429/hess-2017-429-RC2-supplement.pdf

---

## Author Comment (AC1) · 21 Sep 2017

Responses to review comments (Anonymous Referee 1)

General comments: This manuscript investigates the seasonal cycles and trends of water budgets over 18 river basins in the Tibetan Plateau using a wide range of datasets from satellite-observed, land-surface-models simulated, reanalysis and upscaled results of in-situ observations. Prior to seeking the general hydrological features over 18 basins under the Budyko framework, they first assessed the accuracy of six ET products using the water-balance-method derived ET values. They also found that P, Q and ET generally increased in past 30 years in most basins, demonstrating an

overall moistening trend in TP. While a quantitative illustrating the uncertainties in the results is very difficult due to the lack of in-situ observation over this remote area, the authors indeed documented the possible uncertainties in the selected datasets, providing a helpful clue for future research. In my view, this MS describes interesting results, which contributes to advancing our understanding of the hydrological cycle regime over such a hydro-meteorologically important but sparsely-instrumental area. Overall, the MS is nicely structured and presented, and is of a topic that should be of interest to the readers of HESS. I think it is publishable after addressing the comments below. My recommendation is minor revision.

Thanks for the invaluable comments. We have revised the manuscript accordingly (please see the point-to-point responses below) based on your suggestions.

Major comments: I think some analyses over the westly-controlled basins need to be revised because of its special climate patterns compared to other basins. L356-357: "more snow melt contributions" may be due to its special seasonality of precipitation in the westlies-controlled basins rather than its "colder" status.

Thanks for the nice suggestions. We totally agree with you. We have revised the analyses in the westlies-controlled basins as follows (Line 350-354 in the new version), "...For example, in the westerlies-controlled basins, more glaciers developed due to their relatively colder air temperature and special seasonality of precipitation. Therefore, there are more snow melt contributions to total river streamflow with global warming during the period 1983-2006...".

By the way, could you show the annual mean temperature for each basin in Figure 6 to support they are indeed "colder"?

Actually, the annual mean temperature for each basin has been exhibited in Table 1. It showed that the basin-averaged air temperature in the westelies-controlled basins were relatively colder than that in other monsoon-dominated basins.

HESSD
Also, I think the (limited) water availability plays a more important role than the heat stress (i.e. colder status) in leading to a relatively less vegetation over such basins. From Figure 4 c and d, it appears than R2 between ET and NDVI (0.76) is much higher than that between T and NDVI (0.35).

We agree that, thanks. In the revised version, we have revised the related descriptions as follows (Line 338-347 in the new version), "...Overall, from the westerlies-dominant, Indian monsoon-dominant to East Asian monsoon-dominant basins, the annual mean air temperature (-5.68 0.97 oC) and ET (and thus runoff coefficient gradually decreases) increases while the multiyear mean glacier area (and thus the glacier melt normalized by precipitation) gradually decreases (Fig. 4 and Table 2). Moreover, the vegetation status (NDVI range: 0.05 0.43; LAI range: 0.03 0.83) tends to be better. The R2 between basin-averaged NDVI and ET (0.76) is much higher than that between T and NDVI (0.35), which indicates that the water availability plays a more important role than the heat stress (i.e., colder status) over such basins...".

Specific comments: L3: "river basins" may be more appropriate than "rivers"

We have revised the title of this manuscript for "Investigating water budget dynamics in 18 river basins across Tibetan Plateau through multiple datasets" based on the reviewer's suggestion. Thanks.

L67: Most stations are located only in eastern TP and few of them situated in the western part. It would be better if you can highlight such a challenge.

We have highlighted the unevenly distributed pattern in the revised version in Line 68-69 as follows, "...at relatively low elevation regions (most station are located in the eastern TP and few of them situated in the western parts)...".

L71: It seems that "snow depth" is recorded by these stations. I suggest deleting such a term.

Done! Thanks.
L77: labor and/or technical support for maintaining in-situ observation is also a great challenge in addition to the high cost.

We have added these aspects into the sentence in the revised version in Line 77-78 as follows, "...the overall cost, labor and technical support for running the operational sites would be substantial...". Thank you very much.

L419: reword "attributed to the ascending P exceed the increase in PET" as "due to the higher rates of the increase of P than that of PET"

Based on the reviewer's suggestion, we have revised this sentence accordingly in the new version in Line 419-420 as follows, "...the PET/P declined due to the higher rates of the increase of P than that of PET".

L426: change "precipitation" to "rain"?

Changed! Thanks!

L463: The increase of PET/P may be consistent to the changes in moisture flux of TP, as illustrated by Gao et al. (2014). Reference: Gao. Y., Cuo, L., Zhang, Y. 2014. Changes in Moisture Flux over the Tibetan Plateau during 1979–2011 and Possible Mechanisms. J. Clim., doi: 10.1175/JCLI-D-13-00321.1

We totally agree with the reviewer. We have downloaded/read/cited this paper in the revised version as follows (Line 461-463). "...The increased PET/P in Brahmaputra River basin may be consistent with the drying moisture flux in the southeastern TP, as illustrated by Gao et al. (2014)..."

L553: change "; receded at some tributaries" to "with the exception of some tributaries of"

Changed!

L555: revise "a decrease trend" and change "corresponded" to "corresponds"
Done!

L562: change "indicated" to "indicates"

Changed!

L567: delete "under global warming"

Done! Thanks!

---

## Referee Comment (RC3) · Anonymous Referee #3 · 25 Sep 2017

**Review comments for hess-2017-429**

**Main points:**

Tibetan Plateau is a typical data-sparse and high-altitude region. The basin-wide water and energy budgets over plateau are, so far, not well understood due to the lack of in situ observations of the land surface processes. In this manuscript, the authors investigated the general hydrological regime (e.g. seasonal cycle and trend) in 18 basins over plateau through the use of multi-source dataset. On one hand, the in situ data in plateau is extremely sparse. On the other hand, there are considerable global/regional datasets including observation-based, remote sensing retrievals, land surface model simulations and reanalysis/GCM outputs. It is thus a very interesting way to understand the general water budgets in the plateau through integrating the multiple datasets, although there are certain uncertainty inherits from various data.

The topic fit well with the scope of HESS and the manuscript is overall well-written and organized. I also found it is a resubmission. After going through the old version and the corresponding revisions/responses, I think the old manuscript has been significantly improved. The uncertainty is a challenge for multiple dataset based analysis; it usually cannot be easily investigated in the analysis due to the consistency of different datasets (for example, the TRMM and GRACE data are only available after 2000; they are thus difficult to incorporate in the main analysis during 1982-2011). However, the authors have carefully compared the obtained results with some of the existing observation-based studies and discussed the uncertainty issues in Section 3.4 (Table 3 and figure 12 and figure 13). I think it is reasonable. Overall, I do not find major problems with this manuscript and would recommend its publication after minor revision considering the issues rose below.

**Minor points:**

(1) How did you consider the water balance closure in your study?

(2) Line 25: I suggest add "in situ" before "hydro-climatic" to make the sentence more clearly.

(3) Line 26: "the seasonal cycles and trends…"

(4) Line 35-37: This sentence is not clear to me. How about change it for "…past 30 years, except for …Yalong River which were…East Asian Monsoon"?

(5) Line 56-57: "…and their responses to".

(6) Section 2.2.3. I think this paragraph is not useful.

(7) Line 346 and Line 348. I suggest unify the use of " ~ " and "—" between two data throughout the manuscript.

(8) Line 350: It should be Table 2?

(10)Line 536: please delete "of glacier and".

(10) Figure 10: it is difficult to find whether the trend of Q/P in Xining station is positive or negative.

---

## Author Comment (AC2) · 26 Sep 2017

This manuscript investigated the dynamics of water budgets of the 18 river basins over Tibetan Plateau (TP) by multisource datasets including in situ observations, satellite retrievals, reanalysis outputs and land surface model outputs. The actual evapotranspiration was estimated using a water balance-based two-step procedure which considered the changes in basin-scale water storage at the annual scale. Their results show that precipitation is the major contributor to the runoff in TP basins and the weakening East Asian Monsoon mainly affected the increased water budget components. It offers a helpful insight towards understanding the water and energy budgets and sustainability

of water resource management practices in the data-sparse TP region based on the current-existing multi source datasets. Overall, the topic and results of the manuscript are very interesting and meaningful, and fit well with the scope of HESS. It is also well-written and organized. I have also noticed that the manuscript is a resubmission. After reading the review comments and the corresponding responses in the last time, I found the manuscript has been significantly improved. In my opinion, the manuscript could be considered for publication after some minor revisions this time.

Thank you very much for the invaluable comments/suggestions. Based on your suggestions, we have revised the manuscript accordingly (please see the point-to- point responses below).

Specific comments: Some typo errors given below need to be further corrected, for example, 1. Lines 226-228: The sentence "It has been demonstrated cannot be neglected . . .." should be re-written.

We have rewritten it for "It has been demonstrated that  $\Delta S$  cannot be..." in Line 245 in the revised version. Thank you.

2. Lines 294-295, 306, 307 and 307: The unit of statistical indicators should be uniformed, such as "RMSE=8.34 mm/month" in line 294, "RMSE=5.69 mm month-1". The authors should change them in the whole manuscript.

Thank you for your nice suggestions. In the revised manuscript, we have unified them by using the "mm/month".

3. Line 343-352, these sentences should be rewritten to make them more readable.

We have rewritten these sentences as follows (Line 338-347 in the new version) "...Overall, from the westerlies-dominant, Indian monsoon-dominant to East Asian monsoon-dominant basins, the annual mean air temperature (-5.68 0.97 oC) and ET (and thus runoff coefficient gradually decreases) increases while the multiyear mean glacier area (and thus the glacier melt normalized by precipitation) gradually decreases
(Fig. 4 and Table 2). Moreover, the vegetation status (NDVI range: 0.05 0.43; LAI range: 0.03 0.83) tends to be better. The R2 between basin-averaged NDVI and ET (0.76) is much higher than that between T and NDVI (0.35), which indicates that the water availability plays a more important role than the heat stress (i.e., colder status) over such basins..."

4. Lines 346, the use of " " and "-" should also be unified for the entire manuscript. ËĞ

Thank you. We have unified them using the "-"for the entire manuscript in the new version.

5. Line 350, Table 1 or Table 2?

It is Table 2. We have revised it in the new version.

6. Lines 420: "change" should be "changes".

Done!

7. Line 562: "indicates"??

We have changed "indicated" for "indicates" in the revised version. This manuscript investigated the dynamics of water budgets of the 18 river basins over Tibetan Plateau (TP) by multisource datasets including in situ observations, satellite retrievals, reanalysis outputs and land surface model outputs. The actual evapotranspiration was estimated using a water balance-based two-step procedure which considered the changes in basin-scale water storage at the annual scale. Their results show that precipitation is the major contributor to the runoff in TP basins and the weakening East Asian Monsoon mainly affected the increased water budget components. It offers a helpful insight towards understanding the water and energy budgets and sustainability of water resource management practices in the data-sparse TP region based on the currentexisting multi source datasets. Overall, the topic and results of the manuscript are very interesting and meaningful, and fit well with the scope of HESS. It is also well-written and organized. I have also noticed that the manuscript is a resubmission. After read-
ing the review comments and the corresponding responses in the last time, I found the manuscript has been significantly improved. In my opinion, the manuscript could be considered for publication after some minor revisions this time.

Thank you very much for the invaluable comments/suggestions. Based on your suggestions, we have revised the manuscript accordingly (please see the point-to- point responses below).

Specific comments: Some typo errors given below need to be further corrected, for example, 1. Lines 226-228: The sentence "It has been demonstrated cannot be neglected . . .." should be re-written.

We have rewritten it for "It has been demonstrated that  $\Delta S$  cannot be..." in Line 245 in the revised version. Thank you.

2. Lines 294-295, 306, 307 and 307: The unit of statistical indicators should be uniformed, such as "RMSE=8.34 mm/month" in line 294, "RMSE=5.69 mm month-1". The authors should change them in the whole manuscript.

Thank you for your nice suggestions. In the revised manuscript, we have unified them by using the "mm/month".

3. Line 343-352, these sentences should be rewritten to make them more readable.

We have rewritten these sentences as follows (Line 338-347 in the new version) "...Overall, from the westerlies-dominant, Indian monsoon-dominant to East Asian monsoon-dominant basins, the annual mean air temperature (-5.68 0.97 oC) and ET (and thus runoff coefficient gradually decreases) increases while the multiyear mean glacier area (and thus the glacier melt normalized by precipitation) gradually decreases (Fig. 4 and Table 2). Moreover, the vegetation status (NDVI range: 0.05 0.43; LAI range: 0.03 0.83) tends to be better. The R2 between basin-averaged NDVI and ET (0.76) is much higher than that between T and NDVI (0.35), which indicates that the water availability plays a more important role than the heat stress (i.e., colder status)
over such basins ... "

4. Lines 346, the use of " " and "-" should also be unified for the entire manuscript. ËĞ

Thank you. We have unified them using the "-"for the entire manuscript in the new version.

5. Line 350, Table 1 or Table 2?

It is Table 2. We have revised it in the new version.

6. Lines 420: "change" should be "changes".

Done!

7. Line 562: "indicates"??

We have changed "indicated" for "indicates" in the revised version. Thanks.

---

## Author Comment (AC3) · 30 Sep 2017

Responses to review comments (Anonymous Referee 3)

Main points:

Tibetan Plateau is a typical data-sparse and high-altitude region. The basin-wide water and energy budgets over plateau are, so far, not well understood due to the lack of in situ observations of the land surface processes. In this manuscript, the authors investigated the general hydrological regime (e.g. seasonal cycle and trend) in 18 basins over plateau through the use of multi-source dataset. On one hand, the in

situ data in plateau is extremely sparse. On the other hand, there are considerable global/regional datasets including observation-based, remote sensing retrievals, land surface model simulations and reanalysis/GCM outputs. It is thus a very interesting way to understand the general water budgets in the plateau through integrating the multiple datasets, although there are certain uncertainty inherits from various data. The topic fit well with the scope of HESS and the manuscript is overall well-written and organized.

I also found it is a resubmission. After going through the old version and the corresponding revisions/responses, I think the old manuscript has been significantly improved. The uncertainty is a challenge for multiple dataset based analysis; it usually cannot be easily investigated in the analysis due to the consistency of different datasets (for example, the TRMM and GRACE data are only available after 2000; they are thus difficult to incorporate in the main analysis during 1982-2011). However, the authors have carefully compared the obtained results with some of the existing observationbased studies and discussed the uncertainty issues in Section 3.4 (Table 3 and figure 12 and figure 13). I think it is reasonable. Overall, I do not find major problems with this manuscript and would recommend its publication after minor revision considering the issues rose below.

Many thanks for your invaluable comments/suggestions. Based on your suggestions, we have revised the manuscript accordingly (please see the point-to- point responses below).

Minor points: (1) How did you consider the water balance closure in your study?

In this study, we considered basin-scale water balance through the water balance equation (P-Q-ET=D(s), P is observed precipitation, Q is observed streamflow at the basin outlet, ET is basin-wide evapotranspiration and D(s) is water storage change). Because ET cannot directly be observed at the basin-scale, we defined P-Q-D(s) as true ET at the basin scale based on water balance equation when D(s) can be estimated
from GRACE data. We further defined P-Q as biased ET when GRACE data is unavailable. We corrected biased ET by true ET using a water-balance based two-step procedure. The water balance closure is empirically considered during the process of bias-correction in ET calculation.

(2) Line 25: I suggest add "in situ" before "hydro-climatic" to make the sentence more clearly.

Added! Thanks.

(3) Line 26: "the seasonal cycles and trends..."

Done!

(4) Line 35-37: This sentence is not clear to me. How about change it for "...past 30 years, except for ...Yalong River which were...East Asian Monsoon"?

We totally agree with you. In the new version, we have revised this sentence as follows (Line 35-37 in the revised manuscript), "Increased P, ET and Q were found in most TP basins during the past 30 years, except for the upper Yellow River basin and some sub-basins of Yalong River which were mainly affected by the weakening East Asian Monsoon". Thanks.

(5) Line 56-57: "...and their responses to".

Revised!

(6) Section 2.2.3. I think this paragraph is not useful.

We have removed Section 2.2.3 in the new version, Thank you.

(7) Line 346 and Line 348. I suggest unify the use of "" and "—" between two data throughout the manuscript.

We have unified them for "—" throughout the revised manuscript.

(8) Line 350: It should be Table 2?
Yes, revised.

(9)Line 536: please delete "of glacier and".

Deleted, thanks.

(10) Figure 10: it is difficult to find whether the trend of Q/P in Xining station is positive or negative.

The Sen's slope is 0 for Q/P in Xining. It means that the Q/P in Xining station is unchanged.

---

## Author Response (AR1)

**Responses to Editor's comments:**

This manuscript is a re-submission, based on a previous manuscript (https://www.hydrol-earth-syst-sci-discuss.net/hess-2016-624/). The 3 reviewers are relatively satisfied with the revised manuscript, and all recommend minor revision. The authors' responses indicate that they accept the reviewer comments, and indeed the authors have already submitted a revised manuscript addressing all reviewer comments.

I am generally satisfied with the manuscript and the authors' responses to the review comments. I have two editorial suggestions:
**Thanks for the invaluable suggestions. We have revised the manuscript accordingly (*please see the point-to-point responses below*).**

1. A statement to the effect that "Increased P, ET and Q were found in most TP basins during the past 30 years except for the upper Yellow River basin and some sub-basins of Yalong River," appears in several places in the manuscript. I think it needs to be qualified because few of the increases were shown to be statistically significant. The fact that so many of the trend slopes are positive is certainly suggestive of an overall increase, but either more argument is needed to make this point, or additional qualification should be added.

Here I summarise the number of statistically significant increases shown in Figs 8, 10, 11
Westerly-dominated: Fig 8: P 0/3 ; ETwb 0/3; Q 1/3
East-Asian monsoon Fig 10: P 1/9 ; ETwb 3/9; Q 0/9
Indian monsoon Fig 11: P 2/6 ; ETwb 4/6; Q 1/6
Total over all basins: P 3/18; ETwb 7/18; Q 2/18

In total, 12 out of these 54 variables show a statistically significant increase.
**We totally agree with you. In the revised version, we have added more additional qualifications to the expressions of results related to the trends, using more precise words (*i.e., significant/insignificant, markedly/slightly, distinctly/marginally*). We are sure that the improved expressions (*please see the revisions in Section 3.3 and the related parts in Summary and Abstract*) can be more accurate and clear to readers. Thank you very much.**

2. The presentation of the 18 basins in Tables 2,3,4 would be improved if they were separated into the 3 sub-groups used elsewhere in the manuscript.
**Done! Thanks. Please see the revised Tables R1-R3 (*Tables 2, 3 and 4 in the new version*) in the following pages in this file.**

[revised manuscript text omitted]